# Aggregating Algorithm and Axiomatic Loss Aggregation

**Armando J. Cabrera Pacheco**[*]                    *a.cabrera@uni-tuebingen.de*
*University of Tübingen*
*and Tübingen AI Center*
*Tübingen, 72070 Germany*

**Rabanus Derr**[*]                                 *rabanus.derr@uni-tuebingen.de*
*University of Tübingen*
*and Tübingen AI Center*
*Tübingen, 72070 Germany*

**Robert C. Williamson**                            *bob.williamson@uni-tuebingen.de*
*University of Tübingen*
*and Tübingen AI Center*
*Tübingen, 72070 Germany*

**Reviewed on OpenReview:** *https://openreview.net/forum?id=4bUuWtOuDx*

## Abstract

Supervised learning has gone beyond the empirical risk minimization framework. Central to most of these developments is the introduction of more general aggregation functions for losses incurred by the learner. In this paper, we turn towards online learning under expert advice. Via easily justified assumptions we characterize a set of reasonable loss aggregation functions as quasi-sums. Based upon this insight, we suggest how to tailor Vovk's Aggregating Algorithm to these more general aggregation functions. The "change of variables" we propose, let us highlight that "weighting profiles" determine the contribution of each expert to the next prediction according to their loss and the multiplicative structure of the weight updates in the Aggregating Algorithm translates into the additive structure of the loss aggregation in the regret bound. In addition, we suggest that the mixability of the loss function, which is functionally necessary for the Aggregating Algorithm, is intrinsically relative to the log loss, because the standard aggregation of losses in online learning is the sum. Finally, we conceptually and empirically argue that our generalized loss aggregation functions express the attitude of the learner towards losses.

## 1 Introduction

Whether it is framed as collaborative learning, distribution shift or data corruption, machine learning scholarship encountered the necessity to rethink the gold standard of learning theory: expected risk minimization.[1] Central to this paradigm is the minimization of the average loss over a set of instances incurred by the learner. In the named, more recent learning scenarios the average is replaced by other aggregation functionals which take into account the different data sources (Haghtalab et al., 2022, Eq. (1)), the distribution shift (Rahimian and Mehrotra, 2019, Eq. (R0)) or structured noise (Iacovissi et al., 2023). See (Fröhlich and Williamson, 2024) for a nice stratification of reasonable aggregations and an axiomatic approach to loss aggregation in offline learning. However, an axiomatic approach to generalized loss aggregations remained within the borders of offline learning. An analogous development in online, adversarial learning does, to the best of our knowledge, not exist. In this work, we focus on learning under expert advice, where a learner, informed by expert predictions, crafts a prediction which ideally performs as good as the best expert in the

---

[*]Equal contribution.
[1]This list is far from being exhaustive, e.g., fair machine learning (Williamson and Menon, 2019).

group of experts. The performance is measured by the regret which is the sum of losses between learners and experts. The sum as aggregation of the instance-wise losses is a deliberate choice which makes every incurred loss equally important.

There have been efforts made to go beyond expert learners on sum-aggregation. For instance, discounted aggregation of losses (Cesa-Bianchi and Lugosi, 2006, p. 32) down-weights the history of incurred losses of learners and experts (Chernov and Zhdanov, 2010). Or, (strongly) adaptive regret (Hazan and Seshadhri, 2007; Daniely et al., 2015) computes the maximal regret on a fixed-length subintervals of instances (Zhang et al., 2020). All of these regret types share that they change the aggregation function of the losses such that certain instance-wise losses are emphasized.[2] Hence, not every prediction made is equally "important" to good performance, as its relative contribution to the regret differs. The generalization of the aggregation functions is motivated by decaying interest in past performance (discounted regret) or interest in good performance in changing environments ((strongly) adaptive regret) (cf. distribution shift in supervised learning theory).

In contrast, our considered aggregations are mainly motivated by judging the importance of large or small losses. For instance, this might be relevant in scenarios like weather-prediction in which catastrophic mispredictions are extremely undesired, as weather warning systems might to be triggered, but slight mispredictions are not considered to be harmful.

The aggregation function for the losses is part of the optimization objective. Hence, the design of the objective requires an informed choice. Motivated by the context which demand for general aggregations beyond the sum named before, we offer an axiomatic access to the choice of loss aggregations. The axioms construe a starting point for informed and consequence-aware objective design.

**Contributions:** First, we put forward an axiomatic approach to reasonable loss aggregations in online learning, providing a generalized definition of regret. More concretely, under easily justified assumptions the aggregation forms a quasi-sum generated by an appropriate function $u\colon [0, \infty) \longrightarrow [0, \infty)$, given by

$$\mathbf{Q}_n^u(x_1, \ldots, x_n) = u^{-1}\left(\sum_{i=1}^n u(x_i)\right).$$

Based upon this insight, we suggest how to tailor the Aggregating Algorithm to these more general aggregation functions. The Aggregating Algorithm, first suggested by Vovk (1990), solves the learning under expert advice problem. It enjoys nice theoretical guarantees, e.g., a time-independent bound on regret and the recovery of Bayes' updating under an appropriate choice of loss function, while keeping simplicity. The "change of variables" we propose in this work, leads us to several structural insights on the Aggregating Algorithm: (a) We identify a "weighting profile" which determines the contribution of each expert to the next prediction. (b) We argue that the multiplicative structure of the weight updates in the Aggregating Algorithm translates into the additive structure of the loss aggregation in the regret bound. (c) We show that the mixability of the loss function, which is functionally necessary for the Aggregating Algorithm, is intrinsically relative to the log loss, because the standard aggregation of losses in online learning is the sum. Finally, we argue that generalized aggregations express the attitude of the learner towards losses incurred by providing predictions. In particular, we can tune the generator $u$ of the quasi-sum to express the aversity towards extreme losses (convex $u$) or the risk-seeking behavior of a learner (concave $u$). We provide experimental evidence corroborating this statement, which closes the loop by motivating the use of generalized aggregations in the first place.

## 2 Vovk's Aggregating Algorithm

For an intuitive and illustrated introduction to Vovk's Aggregating Algorithm see Appendix A. Here, we describe it in a rigorous way following (Vovk, 1990). We also use this part to set up notation and remark its main properties. A *prediction game* $(\Omega, \Gamma, \Theta, \lambda, \eta)$ consists of the following objects:

---

[2]Note that dynamic regret Zinkevich (2003) does not fit into this picture, as dynamic regret changes the comparison baseline, but not the aggregation functional.

**Sample space** $\Omega$. This is regarded as the set of outcomes from nature. In general, we do not impose any structure on it.

**Decision space** $\Gamma$. This is thought as the *allowed predictions*. $\Gamma$ is a topological space endowed with the $\sigma$-algebra generated by the open sets.

**Parameter space** $\Theta$. We index the *experts* (or *decision strategies*) by $\theta \in \Theta$. We define $(\Theta, \mathcal{F}, \mu)$ as a measure space with $\mu$ being a base measure on the $\sigma$-algebra $\mathcal{F}$ on $\Theta$. For the sake of simplicity we do not write out the base measure $\mu$ in the rest of the paper, e.g., we write the Lebesgue-integral for a measurable $E \in \mathcal{F}$, $\mu(E) = \int_E d\theta$.

**Loss function** $\lambda \colon \Omega \times \Gamma \longrightarrow [0, \infty]$. This gives us a way to measure the quality of the predictions.

**Learning rate** $\eta > 0$. A positive real number, typically as large as possible.

**Example 2.1.** *A widely used loss function in the prediction game is the log-loss $\lambda_{\ln} \colon \Omega \times \Gamma \longrightarrow [0, \infty)$. In particular, when elements in $\Gamma$ are of the form $\gamma \colon \Omega \longrightarrow [0, 1]$, we define it as $\lambda_{\ln}(\omega, \gamma) \coloneqq -\ln(\gamma(\omega))$.*

The prediction game works as follows. At each time $t \in [T] = \{1, ..., T\} \subset \mathbb{N}$,

1. The experts make their predictions. That is, we have a measurable map $\xi_t \colon \Theta \longrightarrow \Gamma$. $\xi_t(\theta)$ is interpreted by the prediction made by the expert $\theta$.

2. The learner (who observes the predictions made by the experts) makes a prediction $\gamma_t \in \Gamma$.

3. Nature chooses the outcome $\omega_t \in \Omega$.

The goal of the learner is to ensure that its *cumulative loss* is as good as the best expert's cumulative loss. In other words, we want to bound the *regret*,

$$\mathbf{R}_T(\theta) \coloneqq \mathbf{L}_T(\text{learner}) - \mathbf{L}_T(\theta) \coloneqq \sum_{t=1}^{T} \lambda(\omega_t, \gamma_t) - \sum_{t=1}^{T} \lambda(\omega_t, \xi_t(\theta)), \quad \forall \theta \in \Theta.$$

Vovk's Aggregating Algorithm (Vovk, 1990; 2001) (see Algorithm 1) gives a bound which is independent of the number of played rounds $T$,

$$\mathbf{L}_T(\text{learner}) - \mathbf{L}_T(\theta) \leq \frac{\ln n}{\eta}, \quad \forall \theta \in \Theta, \tag{1}$$

when the number of experts is finite ($n = |\Theta| < \infty$) and when the game is *mixable*, i.e., we assume the existence of a substitution function $\Sigma$ for given $\lambda$ and $\eta$ such that $\lambda(\omega, \Sigma(\psi)) \leq \psi(\omega)$, for all $w \in \Omega$ and all pseudo-predictions $\psi$ of the form (2) below.[3] As one can easily see, mixability is required for the third step of the Algorithm 1. Since the first two steps are independent of how the substitution concretely looks like, they are sometimes summarized as the *Aggregating Pseudo-Algorithm (APA)*. The separation as well simplifies the analysis of the algorithm. A fundamental property of the AA is that, under some conditions, the updating scheme (Step (1) in AA (Algorithm 1)) is reduced to Bayesian updating (see Appendix A.1).

## 3 Generalizing the loss aggregation in learning under expert advice

The primary goal of the AA, as well as other algorithms solving the learning under expert advice problem, is to bound the summed loss of the learner by the summed loss of each of the experts plus an error term. We now turn towards an approach to loss aggregation from first principles. Instead of presuming the standard summation we provide a list of axioms for loss aggregation functionals which we readily justify (see for example (Grabisch et al., 2009)). Clearly, the choice of axioms is, and should *not* be, universal. We consider our suggested list as a starting point for discussions which can be used for designers of machine

---

[3]For several games, the log-loss (Example 2.1) can be shown to be mixable for $\eta \leq 1$.

---

**Algorithm 1:** Aggregating Algorithm (AA)

---

**Data:** Mixable prediction game $(\Omega, \Gamma, \Theta, \lambda, \eta)$ and prior distribution $P_0$.
**Result:** Predictions $(\gamma_t)_{t \geq 0}$.
**for** *every $t$* **do**

    (1) Update experts' weights $P_t(\theta) := e^{-\eta\lambda(\omega_t, \xi_t(\theta))} P_{t-1}(\theta)$ ;
    (2) Provide pseudo-prediction ,

$$\psi_t(\omega) := \ln_{-\eta} \left[ \int_\Theta e^{-\eta\lambda(\omega_t, \xi_t(\theta))} P_{t-1}^* \, d\theta \right], \text{ with } P_{t-1}^*(\theta) := \frac{P_{t-1}(\theta)}{\int_\Theta P_{t-1}(\theta) \, d\theta}. \tag{2}$$

    (3) Substitute the pseudo-prediction by an allowed prediction $\gamma_t := \Sigma(\psi_t)$ such that
    $\lambda(\omega, \Sigma(\psi_t)) \leq \psi(\omega)$, for all $\omega \in \Omega$;

**end**

---

learning systems to justify the desired objective in a certain context. In particular, the axioms structure the existing options for aggregations. Given an axiomatic characterization of aggregations certain properties of aggregations can be discussed, accepted as being important or neglected as being irrelevant.

Our specific choice of axioms emphasize the special role of quasi-sums as they allow for the switch from additive to multiplicative structure necessary for the AA. The set of axioms is potentially not minimal, in the sense that one axiom can be derived from the others. Furthermore, this list of axioms is definitely not exhaustive. However, it is sufficient to disentangle parts and properties of the AA as stated in the contributions in Section 1.

**Definition 3.1** (Aggregation Functions). *A function $\mathbf{A}: \bigcup_{n \in \mathbb{N}} [0, \infty)^n \longrightarrow [0, \infty)$ is called an* aggregation function. *We write $\mathbf{A}_n(x_1, \ldots, x_n)$ for the aggregation of $n$ instances. Let $x_1, \ldots, x_n, x \in [0, \infty)$. We define the following properties of $\mathbf{A}$.*

(A1) **Continuity.** *We say $\mathbf{A}$ is* continuous *if for every $x_i, i \in [n]$,*

$$\lim_{x_i \longrightarrow x} \mathbf{A}_n(x_1, \ldots, x_i, \ldots, x_n) = \mathbf{A}_n(x_1, \ldots, x, \ldots, x_n).$$

(A2) **Monotonicity.** *We say that $\mathbf{A}$ is* strictly increasing *if for $x_i < x_i'$, $i \in [n]$, we have*

$$\mathbf{A}_n(x_1, \ldots, x_i, \ldots, x_n) < \mathbf{A}_n(x_1, \ldots, x_i', \ldots, x_n).$$

(A3) **Associativity.** *We say that $\mathbf{A}$ is* associative *if for all $x \in [0, \infty)$ and $i \in [n]$ we have*

$$\mathbf{A}_1(x) = x,$$
$$\mathbf{A}_n(x_1, \ldots, x_i, \ldots, x_n) = \mathbf{A}_2(\mathbf{A}_i(x_1, \ldots, x_i), \mathbf{A}_{n-i}(x_{i+1}, \ldots, x_n)).$$

(A4) **Loss compatibility**. *We say that $\mathbf{A}$ is* loss compatible *if $\mathbf{A}(0, ..., 0) = 0$.*

Properties (A1)-(A4) seem, dependent on context, relevant to impose on an aggregation $\mathbf{A}$ of losses since they can be interpreted as follows. If $\mathbf{A}$ is continuous, an infinitesimally small change in a loss will result in an infinitesimal change in the aggregate of the losses. If it is strictly increasing, more loss on any instance give more aggregated loss. If it is associative, it is irrelevant how the losses are grouped together for aggregation. Associativity is arguably the most contested axiom, which is not fulfilled by discounted aggregation or adaptive aggregation (cf. Section 1). Furthermore, associativity does most of the heavy-lifting in the characterization result below. However, note that even though this type of associativity immediately implies that the aggregation function is completely determined by the binary aggregation (Grabisch et al., 2009, p. 33), it does not directly imply commutativity.[4] Loss compatibility means that 0 losses aggregate to 0. It turns out we can fully characterize aggregation functions satisfying the properties above.

---

[4] For instance, consider the aggregation which gives back the last value, in terms of the intrinsic ordering of the input, of all elements.

**Lemma 3.2** (axiomatic Characterization of Loss-Aggregations)**.** *Let* $\mathbf{A}\colon \bigcup_{n\in\mathbb{N}}[0,\infty)^n \longrightarrow [0,\infty)$ *be an aggregation function. Suppose that* $\mathbf{A}$ *satisfies* (A1) - (A4)*. Then, there exists a continuous, strictly increasing function* $u\colon [0,\infty) \longrightarrow [0,\infty)$*, with* $u(0) = 0$*, such that*

$$\mathbf{A}_n(x_1,\ldots,x_n) = u^{-1}\left(\sum_{i=1}^n u(x_i)\right). \tag{3}$$

*We call such aggregations* quasi-sums *generated by u, and we denote them by* $\mathbf{Q}^u$*. (Proof F.1)*

**Example 3.3** (*p*-Norms)**.** *Let* $u(x) = x^p$ *for* $p > 0$*. Then* $\mathbf{Q}^u_n(x_1,\ldots,x_n) = \left(\sum_{i=1}^n x_i^p\right)^{1/p}$*.*

If furthermore $\mathbf{A}$ is positively homogeneous, i.e., scaling the losses scales the aggregation of losses $\mathbf{A}_n(\lambda x_1,\ldots,\lambda x_n) = \lambda \mathbf{A}_n(x_1,\ldots,x_n)$ for all $\lambda > 0$, then $u(x) = x^k$ for some $k \in (0,\infty)$ in (3) (Proof F.1).

**Lemma 3.4** (Quasi-Sum to Quasi-Product)**.** *Let* $\mathbf{Q}^u$ *be a quasi-sum. Then, there exists a continuous, strictly decreasing function* $f\colon [0,\infty) \longrightarrow (0,1]$ *with* $f(0) = 1$*, such that*

$$\mathbf{Q}^u_n(x_1,...,x_n) = g\left(f(x_1) \times ... \times f(x_n)\right), \tag{4}$$

*where* $g\colon (0,1] \longrightarrow [0,\infty)$ *is the inverse of* $f$*.[5] (Proof F.2)*

**Example 3.5.** *Let* $u(x) = x$*, then* $f(x) = e^{-x}$ *and the corresponding aggregation of the form* (4) *is the usual sum:*

$$\mathbf{A}_n(x_1,...,x_n) = g\left(f(x_1)...f(x_n)\right) = \mathbf{L}_n(x_1,...,x_n).$$

**Remark 3.6.** *In the remainder of this work we will sometimes write a quasi-sum* $\mathbf{Q}^u$ *as an aggregation function* $\mathbf{A}$ *of the form* (4) *when convenient. This will be explicitly stated or clear from context.*

Equipped with this characterization of reasonable aggregation functions of losses, it is natural to ask how can we solve the extended learning under expert advice problem: In which way do we have to modify existing online learning algorithms in order to provide regret guarantees such that the aggregated loss of the learner is bound above in terms of the aggregated loss of any expert and some constant error term?

## 3.1 Relation to existing literature

The axiomatization of aggregation functionals has a long history (Grabisch et al., 2009). The set of *idempotent* aggregation functionals is of particular relevance to our work. Idempotent aggregations map an *n*-tuple of constant *c*'s to *c*. Averages and generalizations thereof are often idempotent. Hence, offline learning setups usually fall back to idempotent loss aggregations. Online learning most often focuses on sums, a non-idempotent aggregation functional, to offer more fine-grained statements on the regret guarantees.

The set of quasi-arithmetic means forms the set of idempotent analogues to the set of quasi-sums characterized above (see (Grabisch et al., 2009, Section 6.5.1) for details). Kolmogorov and Castelnuovo (1930) and Nagumo (1930) independently provided an axiomatic characterization of this set which strongly resembles our Lemma 3.2. In (Li et al., 2023) the authors use a subset of those quasi-arithmetic means generated by a family of exponential functions, the entropic risk measures, to extend the standard empirical risk minimization framework.

Note that coherent risk measures (Delbaen, 2002), another set of generalized expectations proven useful to generalize empirical risk minimization (Fröhlich and Williamson, 2024), are idempotent and allow for an axiomatic description. However, an axiomatic description of the non-idempotent analogues of coherent risk measures is still an open question. The existence of such a description would be helpful for the use of such aggregations in online learning.

The axiomatic characterizations of aggregations for loss functions should not be conflated with axiomatic characterizations of decision behavior. Certainly, there are relations between the axiomatic characterization

---

[5]For the sake of readability we neglect the multiplication sign in further expressions.

of expected utility behavior based on the seminal work by Von Neumann and Morgenstern (1953) and the axiomatic characterization of quasi-arithmetic means by Kolmogorov and Castelnuovo (1930) and Nagumo (1930) mentioned before (Muliere and Parmigiani, 1993). However, the goals of the axiomatic statement and hence the axioms are quite different. Again differently, an axiomatic approach to aggregations of experts, not the losses, is put forward in (Neyman and Roughgarden, 2023). The authors use quasi-means to pool forecaster in a low-regret fashion.

Finally, in Section 1 we already hinted towards other generalizations of aggregations in online learning. The mentioned discounted loss aggregation fulfill all axioms listed in Definition 3.1, including positive homogeneity, except for associativity. The aggregation implicitly defined in (strong) adaptive, however, are only loss compatible, positive homogeneous and continuous.

## 4 A change of variables for the AA

It turns out that the answer to the question how to learn under expert advice such that the learner can provide regret guarantees under generalized loss aggregations is relatively straightforward. Via a "change of variables"-trick we obtain regret bounds for generalized aggregations relying on the standard Aggregating Algorithm. Despite its simplicity, we take this observation as a starting point to re-investigate the structure and requirements of the Aggregating Algorithm.

**Corollary 4.1** (Change of variables for the AA). *Let $\mathcal{G} \coloneqq (\Omega, \Gamma, \Theta, \widetilde{\lambda}, \eta)$ be a mixable prediction game with $|\Theta| = n$. Furthermore, we assume $\widetilde{\lambda} = u \circ \lambda$ for some loss function $\lambda$ and continuous, strictly increasing function $u \colon [0, \infty) \longrightarrow [0, \infty)$, with $u(0) = 0$. The AA applied on $\mathcal{G}$ achieves the following regret bound:*

$$\mathbf{Q}_T^u(\text{learner}) \coloneqq \mathbf{Q}_T^u(\lambda(\omega_1, \Sigma(\psi_1))...\lambda(\omega_T, \Sigma(\psi_T))) \leq \mathbf{Q}_2^u\left(\mathbf{Q}_T^u(\theta), u\left(\frac{\ln(n)}{\eta}\right)\right), \tag{5}$$

*for any $\theta \in \Theta$.*

*Proof.* Under the assumptions on $\mathcal{G}$ the AA achieves the following regret bound (Vovk, 1990),

$$\sum_{t=1}^{T} u\left(\lambda(\omega_t, \Sigma(\psi_t))\right) \leq \sum_{t=1}^{T} u\left(\lambda(\omega_t, \xi_t(\theta^*))\right) + \frac{\ln(n)}{\eta},$$

which is equivalent to

$$\sum_{t=1}^{T} u\left(\lambda(\omega_t, \Sigma(\psi_t))\right) \leq u\left(u^{-1}\left(\sum_{t=1}^{T} u\left(\lambda(\omega_t, \xi_t(\theta^*))\right)\right)\right) + u\left(u^{-1}\left(\frac{\ln(n)}{\eta}\right)\right),$$

and, because $u$ is strictly increasing,

$$u^{-1}\left(\sum_{t=1}^{T} u\left(\lambda(\omega_t, \Sigma(\psi_t))\right)\right) \leq u^{-1}\left(u\left(u^{-1}\left(\sum_{t=1}^{T} u\left(\lambda(\omega_t, \xi_t(\theta^*))\right)\right)\right) + u\left(u^{-1}\left(\frac{\ln(n)}{\eta}\right)\right)\right),$$

which is (5). $\qquad\square$

**Remark 4.2.** *It is worth to point out that often one is interested in composing loss functions from the "inside", i.e., reparametrizations (Williamson et al., 2016). Here the transformation is extrinsic, we push the loss curve via $u$ obtaining a "distorted" version of the original loss function $\lambda$.*

The lemma shows that it is a matter of perspective to consider the loss $u \circ \ell$ with standard sum aggregation or the loss $\ell$ with $\mathbf{Q}^u$-aggregation. Obtained in this way, the regret bound seems artificial and the arithmetic juggling unmotivated. Therefore, we shift our focus to understanding what is happening in the background, that is, why can a loss distortion be translated into a change of aggregation. In order do so we detour through the development of an Aggregated Algorithm adapted to Quasi-Sums, which is a reparametrization of the standard Aggregating Algorithm. We learn that:

(a) The Aggregating Algorithm involves a weighting function, by default fixed to be the negative exponential $e^{-x}$, which explains the down- the or up-weighting of experts based on their incurred loss. (Section 4.1)

(b) The multiplicative structure of the weight updates directly translates into the additive structure of the loss aggregation. (Section 4.2)

(c) The fundamentality of the log loss in the definition of mixability required for the Aggregating Algorithm is an artifact of the standard use of summation for loss aggregation. (Section 4.3)

Let $\mathcal{G} := (\Omega, \Gamma, \Theta, \widetilde{\lambda}, \eta)$ be a prediction game. In the following, we step-by-step go through the Aggregating Algorithm under the premise that the loss function $\widetilde{\lambda}$ is *decomposable*, i.e., $\widetilde{\lambda} = u \circ \lambda$, with $\lambda$ being a loss function and $u \colon [0, \infty) \longrightarrow [0, \infty)$ being a continuous, strictly increasing function with $u(0) = 0$.

## 4.1 The Weight Updates

Step (1) in Algorithm 1 is the updating of the experts' weights. For the moment, we set the learning rate $\eta = 1$. Under the assumption of decomposability of $\widetilde{\lambda}$, i.e., $\widetilde{\lambda} = u \circ \lambda$, the update can be written as,

$$P_t(\theta) := e^{-\widetilde{\lambda}(\omega_t, \xi_t(\theta))} P_{t-1}(\theta) = f(\lambda(\omega_t, \xi_t(\theta))) P_{t-1}(\theta),$$

for $f(x) := e^{-u(x)}$. In particular, the function $f$ allows for the interpretation as being the profile to judge the experts. It determines how much the expert $\theta$ contributes to the next pseudo-prediction (Step 2) depending on the incurred loss in the current time step. For this reason, we call $f$ *weighting profile*. Weighting profiles fulfill three simple requirements directly derived from the properties of $u$ (cf. Lemma F.2).

**Definition 4.3** (Weighting Profile)**.** *We say that a continuous $f \colon [0, \infty) \longrightarrow (0, 1]$ is a weighting profile if (a) $f(0) = 1$, (b) $f$ is strictly decreasing, and (c) $\lim_{x \to \infty} f(x) = 0$.*

However, the properties of $f$ can be justified independently of the properties of $u$. The normalization $f(0) = 1$ means that weights should be positive and bounded from above by 1. The expert incurring 0 loss should get assigned full weight, while $f$ being strictly decreasing implies that the higher the loss the less weight should be put on the expert. The limiting behavior of $f$ says that an expert which incurs extremely large losses should be punished by getting down-weighted to 0. Note that most of our following statements are framed for a choice of weighting profile $f$ instead for the tantamount choice of $u$. Hence, the change of aggregation $\mathbf{Q}^u$ amounts to the change of weighting profile.

To illustrate Table 1 provides a short list of potential aggregation functions, their corresponding function $u$ (see Section 5.1) and weighting profile. Note that we use the term focal aggregation in the table to emphasize that the composition of the corresponding $u$ with the log-loss recovers the often used focal loss (with $\gamma = 2$) (Lin et al., 2017).

For a fixed loss function we can analyze and compare weighting profiles for different aggregation functions. For instance (see Table 1), observe that for $L_p$-norm aggregations switching the value of $p$ from less than 1 to strictly bigger than 1 drastically changes the shape of the weighting profile. Let us shortly compare the $L_2$-norm and sum. For the sake of simplicity, we focus on the arbitrary learning rate $\eta = 0.5$. The weighting profile corresponding to the $L_2$-norm punishes higher losses stronger than smaller losses compared to the sum. This, however, comes with the cost that for small losses the $L_2$ weighting profile does not finely distinguish between good and even better experts. Both of them get nearly updated with the same weight, in contrast to the sum.

Crucially, comparisons of weighting profiles require to fix a loss a priori. The domain and distribution of the loss values themselves strongly interact with the choice of aggregation. For instance, the Brier score only provides values between 0 and 1. Thus, the weighting profile beyond 1 on the $x$-axis is irrelevant for the comparison. Hence, conclusive interpretations are only possible for fixed losses and different aggregations. Note that mixability might not be maintained for arbitrary combinations of losses and weighting profiles (Section B). For this reason, we go beyond this qualitative interpretation of the effect of aggregations and

Table 1: Aggregations, their corresponding function $u$ and weighting profiles for different learning rates (lightseagreen: $\eta = 0.001$, limegreen: $\eta = 0.5$, mediumseagreen: $\eta = 1$, seagreen: $\eta = 2$, darkgreen: $\eta = 10$, darkslategray: $\eta = 100$).

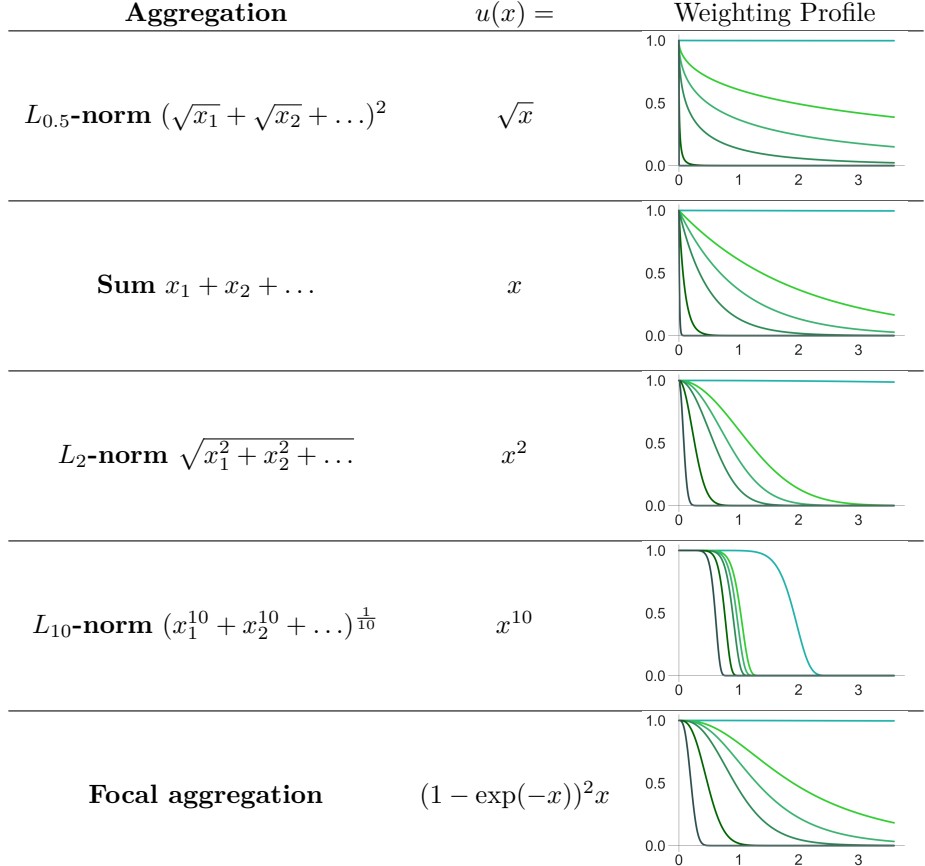

| Aggregation | $u(x) =$ | Weighting Profile |
|---|---|---|
| $L_{0.5}$-**norm** $(\sqrt{x_1} + \sqrt{x_2} + \ldots)^2$ | $\sqrt{x}$ | |
| **Sum** $x_1 + x_2 + \ldots$ | $x$ | |
| $L_2$-**norm** $\sqrt{x_1^2 + x_2^2 + \ldots}$ | $x^2$ | |
| $L_{10}$-**norm** $(x_1^{10} + x_2^{10} + \ldots)^{\frac{1}{10}}$ | $x^{10}$ | |
| **Focal aggregation** | $(1 - \exp(-x))^2 x$ | |

illustrate via an experiment how the aggregation changes the performance of the Aggregating Algorithm in Section 5.2.

The properties of a weighting profile $f$ imply the existence of a continuous inverse, which we denote by $g \colon (0,1] \longrightarrow [0, \infty)$, such that (a) $g(1) = 0$, (b) $g$ is strictly decreasing, and (c) $\lim_{x \to 0^+} g(x) = \infty$. Finally, note that when choosing $\lambda(x) = g(x)$ in the experts' weight updates one recovers Bayes' updating much as described in detail in (Vovk, 2001, Section 2.2) for $\lambda(x) = -\log x$.

## 4.2 The Pseudo-Prediction

Different to the exponential weight algorithm (cf. (Cesa-Bianchi and Lugosi, 2006, p. 14)), which shares the weight updates, the Aggregating Algorithm additional involves a pseudo-prediction step, which is then used to derive the actual prediction. For $\eta = 1$, $\widetilde{\lambda} = u \circ \lambda$ and $f(x) := e^{-u(x)}$ as above, the pseudo-prediction (2) writes as,

$$\psi_t(\omega) = \ln_{e^{-1}} \left( \int_\Theta f(\lambda(\omega, \xi_t(\theta))) P_{t-1}^*(\theta) \, d\theta \right), \quad P_{t-1}^*(\theta) := \frac{P_{t-1}(\theta)}{\int_\Theta P_{t-1}(\theta) \, d\theta}.$$

For notational convenience, we introduce the following slight variant,

$$\psi_t^f(\omega) := u^{-1}(\psi_t(\omega)) = g \left( \int_\Theta f(\lambda(\omega, \xi_t(\theta))) P_{t-1}^*(\theta) \, d\theta \right), \tag{6}$$

where $g$ is the inverse of $f$ as described above. For the sake of readability, we sometimes do not explicitly highlight the dependence of $\psi$ on $f$ in every instance. However, we use the $\psi^f$ notation in those cases where the dependency on $f$ is not clear or particularly important.

The pseudo-prediction allows for several interpretations:

**Normalizing Factor** At each step $t$ we define the $\Omega$-dependent family of measures on $\Theta$, given by

$$p_t(\theta; \omega) = f(\lambda(\omega, \xi_t(\theta))) f(\psi_t^f(\omega))^{-1} P_{t-1}^*(\theta),$$

where $P_{t-1}^*$ is the normalization of $P_{t-1}$. Here, we do not specify $\psi_t^f \colon \Omega \longrightarrow [0, \infty)$. Imposing $p_t(\theta; \omega)$ to be a probability distribution on $\Theta$ yields to the pseudo-prediction given in (6). Hence, we can interpret $\psi_t^f$ as the normalizing factor of $p_t(\theta; \omega)$.

**Loss Mapping** Suppose that $\omega_t \in \Omega$ is revealed by nature. Then

$$\psi_t^f(\omega_t) = g\left[\int_\Theta f(\lambda(\omega_t, \xi_t(\theta))) P_{t-1}^*(\theta)\, d\theta\right].$$

If $\lambda(\omega_t, \xi_t(\theta)) \gg 1$ for all $\theta \in \Theta$, by properties of the weighting profile $f$, the value of $\psi_t^f(\omega_t)$ will be very large. On the other hand, if $\lambda(\omega_t, \xi_t(\theta)) \approx 0$, for all $\theta \in \Theta$, then its value will be close to 0. In this sense, we can interpret $\psi_t^f(\omega_t)$ as the loss incurred by pseudo-prediction $\psi_t^f$ when $\omega_t$ is observed (cf. (Vovk, 2001, Section 2.1)).

Concluding, we summarize the reparametrization of the first two steps of the Aggregating Algorithm in Algorithm 2. Leaning on Vovk (2001)'s naming convention we call it the Aggregating Pseudo-Algorithm for Quasi-Sums (APA-QS). Given the loss interpretation of the pseudo-prediction, we can prove the following Lemma 4.4. Note that we consider the aggregation function $\mathbf{A} \colon \bigcup_{n \in \mathbb{N}} [0, \infty)^n \longrightarrow [0, \infty)$ given by

$$\mathbf{A}_n(x_1, ..., x_n) := g(f(x_1) f(x_2) ... f(x_n)). \tag{7}$$

for a fixed weighting profile $f$ (and hence its inverse $g$). Recall that using (4) $\mathbf{A}$ can be expressed as a quasi-sum $\mathbf{Q}^u$ via the relation $f(x) = e^{-u(x)}$.

---

**Algorithm 2:** Aggregating Pseudo-Algorithm for Quasi-Sums (APA-QS)

**Data:** Prediction game $(\Omega, \Gamma, \Theta, \lambda, \eta)$ and prior distribution $P_0$.
**Result:** Predictions $(\gamma_t)$.
**for** *every $t$* **do**
> (1) Update experts' weights $P_t(\theta) := f(\lambda(\omega_t, \xi_t(\theta))) P_{t-1}(\theta)$ ;
> (2) Provide pseudo-prediction with $P_{t-1}^*(\theta) := \frac{P_{t-1}(\theta)}{\int_\Theta P_{t-1}(\theta)\, d\theta}$,
>
> $$\psi_t^f(\omega) = g\left[\int_\Theta f(\lambda(\omega, \xi_t(\theta))) P_{t-1}^*(\theta)\, d\theta\right]; \tag{8}$$

**end**

---

**Lemma 4.4** (Bound for APA-Loss)**.** *Let $f \colon [0, \infty) \longrightarrow \mathbb{R}$ be a weighting profile and $g$ its inverse. Then*

$$\mathbf{A}_T(\mathrm{APA}(P_0)) := \mathbf{A}_T\left(\psi_1^f(\omega_1), \psi_2^f(\omega_2), ..., \psi_T^f(\omega_T)\right) = g\left[\int_\Theta f(\mathbf{A}_T(\theta)) P_0(\theta)\, d\theta\right].$$

*Moreover, when $|\Theta| = n$ and $P_0$ is the uniform probability distribution with weights $1/n$,*

$$\mathbf{A}_T(\mathrm{APA}(P_0)) \leq \mathbf{A}_2\left(\mathbf{A}_T(\theta^*), g\left(n^{-1}\right)\right), \tag{9}$$

*for any expert $\theta^* \in \Theta$.(Proof F.3)*

The statement follows from yet another "change of variables" for Lemma 1 in (Vovk, 2001). However, for didactic reasons we include a full proof from scratch in the appendix (Lemma F.3). In particular, the proof reveals that the multiplicative structure of the weight updates directly translates into the multiplicative structure of $\mathbf{A}_T$. This gives rise to the additive structure by the exponential relation in (4).

Note that incorporating a learning rate $\eta > 0$ in the APA (as in (Vovk, 2001)) amounts to set $f(x) = e^{-\eta x}$.

**Corollary 4.5.** *Let $f\colon [0, \infty) \longrightarrow \mathbb{R}$ be a weighting profile. Let $\eta \in (0, \infty)$ be a learning rate and define $g_\eta(x) := (f(x)^\eta)^{-1}$. When $|\Theta| = n$ and $P_0$ is the uniform probability distribution with weights $1/n$,*

$$\mathbf{A}_T(\mathrm{APA}(P_0), \eta) = \mathbf{A}_T(\mathrm{APA}(P_0)) \leq \mathbf{A}_2\left(\mathbf{A}_T(\theta^*), g_\eta\left(n^{-1}\right)\right), \tag{10}$$

*for any expert $\theta^* \in \Theta$.(Proof F.4)*

Finally, notice that by setting $f(x) = e^{-x}$ in Corollary 4.5 we recover the bound found by Vovk (1990).

## 4.3 The Substitution Step

As a last step the Aggregating Algorithm derives an actual prediction from the pseudo-prediction. This is achieved by using a substitution function $\Sigma$. The substitution function maps a pseudo-prediction to an allowed prediction such that the loss of the allowed prediction is smaller than the value of the pseudo-prediction evaluated in every possible outcome. More formally, a substitution function $\Sigma$ makes the following inequality hold, $\widetilde{\lambda}(\omega, \Sigma(\psi_t)) \leq \psi_t(\omega)$ for all $\omega \in \Omega$ and all pseudo-predictions $\psi_t$. The existence of such a substitution function is guaranteed by the *mixability* of the game $\mathcal{G} := (\Omega, \Gamma, \Theta, \widetilde{\lambda}, \eta)$ (see Section 2). Concretely, Vovk (2001) suggests the substitution function $\Sigma(\psi) \in \arg\min_{\gamma \in \Gamma} \sup_{\omega \in \Omega} \widetilde{\lambda}(\omega, \gamma)/\psi(\omega)$.[6] Since $\widetilde{\lambda}$ decomposes, we apply the change of variables as well to the definition of mixability. To this end, we introduce $\mathcal{P}(\lambda, f)$ as the set of all pseudo-predictions of the form,

$$\psi^f(\omega) = g\left[\int_\Theta f(\lambda(\omega, \xi_t(\theta))) P_{t-1}^*(\theta)\, d\theta\right] = g\left[\int_\Gamma f(\lambda(\omega, \gamma)) Q(\gamma)\, d\gamma\right], \tag{11}$$

for some distribution $Q$ on $\Gamma$.

**Definition 4.6** (($f, \eta$)-Mixability)**.** *Let $(\Omega, \Gamma, \Theta, \lambda, \eta)$ be a prediction game. Let $f\colon [0, \infty) \longrightarrow [0, 1]$ be a weighting profile and consider pseudo-predictions $\psi^f \in \mathcal{P}(\lambda, f_\eta)$ given by (11). We call $(\Omega, \Gamma, \Theta, \lambda, \eta)$ $(f, \eta)$-mixable there exists a substitution function $\Sigma$ such that*

$$\lambda(\omega, \Sigma(\psi^f)) \leq \psi^f(\omega), \tag{12}$$

*for all $\omega \in \Omega$. If the game is $(f, \eta)$-mixable for some $\eta$, we say the game is $f$-mixable.*

**Lemma 4.7** ($f$-Mixability is Mixability of Composite Loss)**.** *A loss $\lambda$ is $(f, \eta)$-mixable if and only if $\widetilde{\lambda} = u \circ \lambda$ for $u(x) := -\ln f(x)$ is $\eta$-mixable. (Proof F.5)*

As noted by Erven et al. (2011) and Cabrera Pacheco and Williamson (2023), the definition of mixability is a comparison of the loss function $\lambda$ and the learning rate $\eta$ with the log-loss. Our $(f, \eta)$-mixability emphasizes the relativity to some function $f$. Note that if $f(x) = e^{-x} = (-\log x)^{-1}$, the definition directly reduces to the original definition.

Given the prediction game $(\Omega, \Gamma, \Theta, \lambda, \eta)$ is $(f, \eta)$ (respectively the prediction game $(\Omega, \Gamma, \Theta, \widetilde{\lambda}, \eta)$ is $\eta$-mixable), the substitution steps completes the Aggregating Algorithm. We finally obtain Corollary 4.1 in a slightly different, but equivalent, formulation.

**Corollary 4.8.** *Let $f\colon [0, \infty) \longrightarrow \mathbb{R}$ be a weighting profile. Let $\eta \in (0, \infty)$ be a learning rate. When $(\Omega, \Gamma, \Theta, \lambda, \eta)$ is $(f, \eta)$-mixable, $|\Theta| = n$ and $P_0$ is the uniform probability distribution with weights $1/n$,*

$$\mathbf{A}_T(\text{learner}) := \mathbf{A}_T(\lambda(\omega_1, \Sigma(\psi_1^f)), ..., \lambda(\omega_T, \Sigma(\psi_T^f))) \leq \mathbf{A}_2\left(\mathbf{A}_T(\theta^*), g_\eta\left(n^{-1}\right)\right), \tag{13}$$

*for any expert $\theta^* \in \Theta$.*

---

[6]When proper, mixable loss functions are considered in class probability estimation, Kamalaruban et al. (2015) argue that antipolar losses constitute another universal substitution function.

**Remark 4.9.** *The substitution function is a computational bottleneck which, in particular in high-dimensional prediction tasks, might lead to suboptimal time performance. We haven't found literature to address this shortcoming.*

**Remark 4.10.** *Vovk (2015) argues that the log-loss is in a particular way* fundamental. *As Cabrera Pacheco and Williamson (2023) have shown, both Vovk's fundamentality and mixability (in the sufficiently differentiable case) are equivalent to a curvature comparison between a given loss and the log-loss. Definition 4.6 and the analysis of the AA with more general aggregation functions, emphasize that the choice of aggregation is tightly intertwined with the definition of mixability. Particularly, standard sum aggregation corresponds to standard mixability. Hence, the fundamentality of the log-loss is a particularity of the standard sum aggregation and does not imply that the log-loss is fundamental for other aggregations. However, Corollary 4.1 suggest a connection back to a type fundamentality of log-loss via u, which can potentially be described geometrically. We will not go deeper into this observation here.*

The corollary requires the prediction game to be $(f, \eta)$-mixable. However, it turns out that we can generalize the bound on the aggregated loss of the learner to non-mixable losses.

### 4.3.1 The Mixability Constant for Non-Mixable Losses

To slacken the requirement of mixability (Vovk, 2001) introduced the *mixability constant*. In our symbols,

$$c(f) := \inf\{c \in \mathbb{R} \,|\, \forall \psi^f \in \mathcal{P}(\lambda, f), \, \exists \gamma \in \Gamma, \, \forall \omega, \, f(\lambda(\omega, \gamma)) \geq f(\psi^f(\omega))^c\}, \tag{14}$$

and set $\inf\{\varnothing\} := \infty$. Note that $c(f) \geq 1$ (Lemma C.2). Our definition of the mixability constant is equivalent to Vovk (2001)'s definition. This is easy to see, since $f(\lambda(\omega, \gamma)) \geq f(\psi^f(\omega))^c \Leftrightarrow \widetilde{\lambda}(\omega, \gamma) \geq c\psi(\omega)$ (cf. Proof F.5). When the infimum in (14) is attained, a substitution function $\Sigma$ (which also depends on $\eta$ and $\lambda$) exists and satisfies

$$f(\lambda(\omega, \Sigma(\psi))) \geq f(\psi(\omega))^{c(f)}, \tag{15}$$

for all $\omega \in \Omega$.

**Remark 4.11.** *Note that if we fix f and consider $f_\eta(x) = f(x)^\eta$, where $\eta > 0$ is the learning rate, then we can consider the constant $c(f_\eta)$ in (11) to depend only on $\eta$. When we do this we simply denote it by $c(\eta)$.*

With the mixability constant at hand one can provide a generalized bound on the aggregated loss of the learner to non-mixable losses (Theorem C.3), which is analogous to (Vovk, 1990, Eq. (15)). Even the optimality result for the AA (Vovk, 1995), i.e., the constants in this Theorem C.3 cannot be undercut by any other prediction algorithm, translates to our reparametrization and even slightly generalizes the former statement (Section C).

## 5 How aggregation changes prediction

In the previous sections we have argued that the Aggregating Algorithm is generally applicable for losses interacting nicely with "reasonable" aggregation functions. However, it is still unclear how the aggregation influences the actual predictions made by the Aggregating Algorithm.

We qualitatively approach this question in two ways. First, we propose to interpret the generator functions of aggregations as utility functions. Then, we illustrate in an experiment that aggregations can actually express the forecaster's attitude towards losses.

### 5.1 Aggregation and utility of losses

Aggregation functions for losses are, under mild conditions, quasi-sums $\mathbf{Q}^u$ (cf. Lemma 3.2). On one hand, as we have shown, the $u$-quasi-sum of losses $\lambda$ in the regret bound is tantamount to summing up distorted losses $u \circ \lambda$ (Corollary 4.1). On the other hand, *u can be interpreted as* the negative utility function for the losses $\lambda$ of the predictors. It expresses the dis-satisfaction of the learner to incur certain losses. Therefore,

whether we talk about a certain choice of negative utility of losses or whether we talk about $u$-quasi-sums as aggregation functional does not make a difference. For an illustrated, comparative example see Appendix D.

More generally, it is true that risk-avoider prefer convex $u$, i.e., high losses are up-valued, low losses are down-valued. In contrast, risk-taker consider concave $u$, which means that low losses are up-valued and high losses are down-valued (cf. (Winkler and Murphy, 1970), concavity and convexity are switched therein for reasons of sign flip). Hence, we can conclude: **the type of aggregation captures the attitude of the forecaster towards losses.**

### 5.2 Weather prediction via Aggregating Algorithm

The preceding discussions suggest that aggregation, weighting profile and utility are essentially different facets of the same object. We earlier asked, how does the type of aggregation change the behavior of the Aggregating Algorithm? The proposed interpretations as weighting profile and utility lead us to the following three hypothesis, which we substantiate by a real-world data prediction experiment.

**H-(a)** Convex additive generators express the aversion towards extreme losses.

**H-(b)** Concave additive generators express the risk-loving behavior in terms of accepting extreme losses, but seeking for the "perfect" predictor.

**H-(c)** Focal aggregation expresses aversion towards extreme losses.

We provide the log-loss (sometimes called cross-entropy loss) profile of several applications of the AA-QS for different aggregations on a sequential weather classification task. We use the Aggregating Algorithm to aggregate probabilistic predictions of 9 simple classification algorithms. The AA-QS was implemented using the log-loss. The task is detailed in Appendix E. Table 2 summarizes our findings for the data collection from the Zugspitze (Germany). See Appendix E.1 for further experiments. The loss histograms particularly reveal the difference between convex and concave utility function. Concave utility functions (i.e., $L_{0.5}$ and Sum) lead to a high number of predictions with extremely small loss values, with the downside that some predictions incur a high loss. Convex utility functions (i.e., $L_2$ and Sum) damp the tails of high losses, i.e., only few predictions with high losses are made. On the other hand, there are many predictions made incurring small but suboptimal loss. In this realm, $L_{10}$ is the most extreme example in which many sub-optimal predictions are made, but the tails are banned. The focal aggregation leads to behavior largely similar to the one induced by $L_2$-aggregation. All three hypothesis about the change of prediction given a certain aggregation can be substantiated in this explorative study. An exhaustive experimental study would be required to provide more conclusive statements. Nevertheless, there is a conceptual problem in comparing different aggregation schemes on a given problem and asking which one is "better". We are not proposing different algorithms for a given objective. We actually change the objective.

## 6 Conclusion

In this paper, we put forward a general, axiomatic approach to loss aggregation in online learning. Analogous to the development in offline learning, but differently motivated, we show that a set of reasonable aggregations in online learning is characterized by the set of quasi-sums. It turns out that the AA can be adapted to deal with those general aggregations. Not only can we transfer the nice theoretical properties of the AA to its modified variant, we also provide experimental evidence that the choice of general aggregation determines the extreme-loss seeking or extreme-loss averse behavior of the AA. Hence, we span up a new dimension of choice, for which we think that the modified AA is just a starting point. We believe that similar modifications can be provided for other online learning algorithms such as weighted majority algorithm (Littlestone and Warmuth, 1994), follow the regularized leader (Abernethy et al., 2009), or follow the perturbed leader (Kalai and Vempala, 2005).

Table 2: Aggregations, and their corresponding utility function $u$. The loss histogram shows the number of predictions (y-axis) incurring the loss (x-axis) in a certain bin. The blue line depicts the average loss.

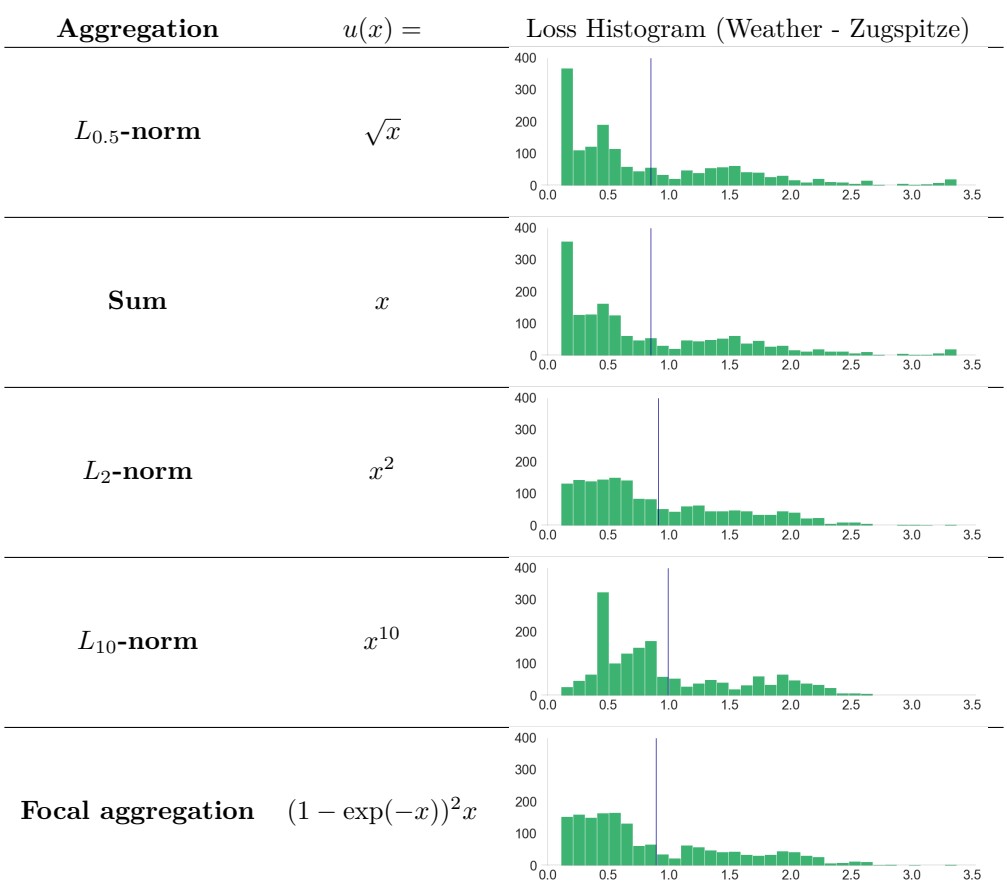

| Aggregation | $u(x) =$ | Loss Histogram (Weather - Zugspitze) |
|---|---|---|
| $L_{0.5}$-norm | $\sqrt{x}$ | |
| Sum | $x$ | |
| $L_2$-norm | $x^2$ | |
| $L_{10}$-norm | $x^{10}$ | |
| Focal aggregation | $(1 - \exp(-x))^2 x$ | |

### 6.1 Broader impact and position statement

The generality of the learning under expert advice setting allows for the deployment of the adapted Aggregating Algorithm in a variety of settings, which do not exclude any malicious nor benevolent use. Note that the type of aggregation creates another choice parameter which, depending on the deployment context, might call for a participatory, democratic approach to the determination of the used quasi-sum. This contextualization requires further studies. We are convinced that our socialization has shaped our method and approach to research. We might have been ignorant to aspects of our work which against other socio-cultural backgrounds might miss or need reframing.

## 7 Acknowledgements

The authors appreciate and thank the International Max Planck Research School for Intelligent System (IMPRS-IS) for supporting the second author. Part of this project was conducted when the second author was visiting Aaron Roth at the University of Pennsylvania. The stay was supported by a fellowship of the German Academic Exchange Service (DAAD). This work was funded in part by the Deutsche Forschungsgemeinschaft (DFG, German Research Foundation) under Germany's Excellence Strategy — EXC number 2064/1 — Project number 390727645; it was also supported by the German Federal Ministry of Education and Research (BMBF): Tübingen AI Center.

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

# A    A simple introduction to Aggregating Algorithm

The Aggregating Algorithm is a relatively straightforward algorithm with strong theoretical guarantees. In order to motivate our theoretical development in this paper and to provide a low-threshold introduction to this algorithm, we first lead the reader through a sensibly simplified scenario of learning under expert advice.

Consider a binary class probability estimation task on the outcome space $\Omega := \{0, 1\}$. We repeatedly see 2 experts $\Theta := \{\theta_1, \theta_2\}$ predicting the probability that the next outcome will be 1, i.e., in every round $t \in [T] := \{1, \ldots, T\}$ each expert $\theta \in \Theta$ provides a prediction $\xi_t(\theta) \in [0, 1]$. After the predictions are given, the learner (who has seen the experts' predictions) has to commit to a prediction $\gamma_t \in [0, 1]$ as well. Then, nature reveals an outcome $\omega_t \in \{0, 1\}$. We measure the quality of the experts' and learner's prediction by the log-loss, that is, $\lambda(s, \omega_t) = [\![\omega_t = 0]\!] (- \ln(s)) + [\![\omega_t = 1]\!] (- \ln(1 - s))$, for $s \in [0, 1]$ ($s$ here refers to the prediction made either by the expert or the learner). Note that the set of possible predictions can be regarded as the probability simplex $s \mapsto (s, 1 - s)$ for $s \in [0, 1]$, cf. top-left in Figure 1. Moreover, we can interpret the log-loss as an embedding of the simplex into $\mathbb{R}^2$, i.e., $(s, 1 - s) \mapsto (- \ln(s), - \ln(1 - s))$, see top-right in Figure 1.

Now, the Aggregating Algorithm, as a *learner*, uses the embedding of the simplex into $\mathbb{R}^2$ and *exponentiates* it by the *exponential mapping* $(\lambda_0, \lambda_1) \mapsto (e^{-\eta \lambda_0}, e^{-\eta \lambda_1})$, where $\eta > 0$ is called the learning rate. Figure 1 right-top to bottom illustrates this step. In particular, the predictions of the experts can be traced through both mappings, the log-loss and the exponential mapping. The Aggregating Algorithm then forms a convex combination of the mapped experts' predictions $\psi_t$. The weights on how much each experts' prediction contributes to this mixture are based on a generalized Bayes' updating (see Section A.1). The updating puts more weight on the experts which performed well in the past. As Figure 1 (orange-brown) illustrates, the obtained convex combination is not necessarily on the exponentiated embedding of the simplex anymore. We later call it a *pseudo-prediction* for this reason. Pseudo-predictions have nice theoretical properties (see Section 4.2), but they are not helpful as predictions, since they are not in the prediction space. That is why the Aggregating Algorithm requires a characteristical step: the substitution $\Sigma$ of the pseudo-prediction by an actual prediction with similar theoretical properties. This substitution $\Sigma$ is intuitively a projection of the pseudo-prediction $\psi$ to the exponentiated embedded simplex (cf. Figure 1 darkgreen). Crucially, a property called "mixability" (see Definition 4.6) guarantees that the pseudo-prediction lies bottom-left to the exponentiated embedded simplex. The obtained actual predictions guarantee that the accumulated loss of the Aggregating Algorithm (learner), i.e., $\sum_{t=1}^T \lambda(\gamma_t, \omega_t)$ with $\gamma_t = \Sigma(\psi_t)$ is always smaller than the accumulated loss of any expert, i.e., $\sum_{t=1}^T \lambda(\xi_t(\theta), \omega_t)$ for all $\theta \in \Theta$, up to a constant $C$. In other words, the Aggregating Algorithm's regret is bounded above by a constant, independent of the number of played rounds:

$$\sum_{t=1}^T \lambda(\gamma_t, \omega_t) - \sum_{t=1}^T \lambda(\xi_t(\theta), \omega_t) \leq C, \forall \theta \in \Theta.$$

## A.1    Bayes' updating for weights

A fundamental property of the AA is that, under some conditions, the updating scheme (Step (1) in AA (Algorithm 1)) is reduced to Bayesian updating (Vovk, 2001, Section 2.2). More precisely, make the following

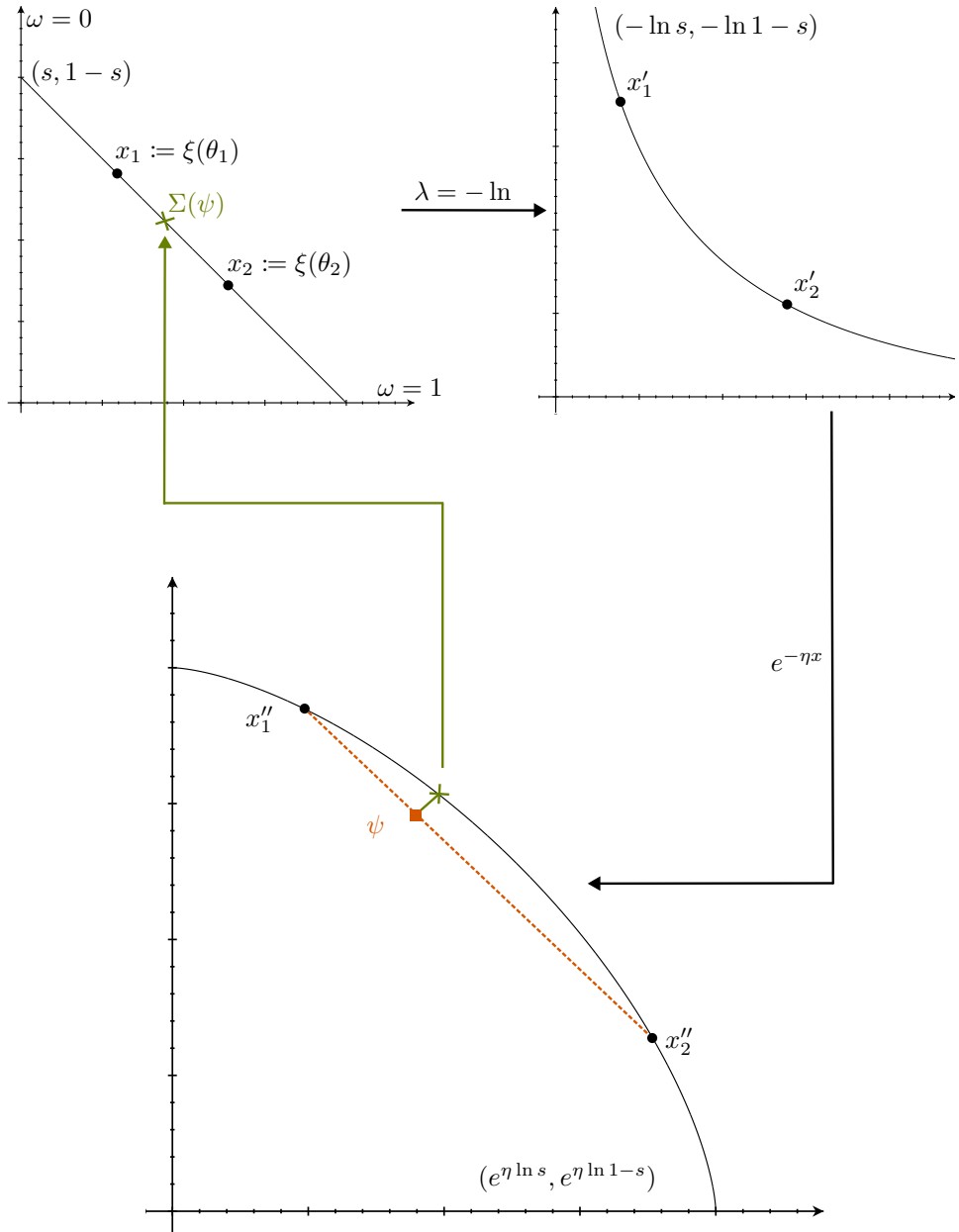

Figure 1: Graphical Summary of the Steps in the Aggregating Algorithm. Experts $\theta_1$ and $\theta_2$ provide predictions $\xi(\theta_1)$ and $\xi(\theta_2)$, respectively, which are placed in the simplex (top-left) as $x_1 := (\xi(\theta_1), 1 - \xi(\theta_1))$ and $x_2 := (\xi(\theta_2), 1 - \xi(\theta_2))$ via $s \mapsto (s, 1 - s)$. The log-loss embeds the simplex as a curve in $\mathbb{R}^2$ (top-right), i.e., $s \mapsto -\ln s$ is applied coordinate-wise and maps $x_1$ and $x_2$ to $x_1'$ and $x_2'$. Then, the exponential mapping projects them into $[0, 1]^2$. The Aggregating Algorithm forms a convex combination $\psi$ of the projected predictions $x_1''$ and $x_2''$ based on weights updated by a Bayesian-type formula (orange-brown), called a pseudo-prediction , which is substituted back to the simplex via a substitution function $\Sigma$ (darkgreen).

choices: (a) $\Omega$ is finite, (b) $\Gamma = \Delta(\Omega)$, the set of all probability measures on $\Omega$, (c) loss function is the log-loss $\lambda_{\ln}$ and (d) the learning rate $\eta = 1$.

For an expert $\theta$ whose prediction at time $t$ is $\xi_t(\theta) \in \Delta(\Omega)$, we write $\xi_t(\theta)(\omega_t)$ as the probability of seeing $\omega_t$ forecasted by $\theta$, who has observed all data points up to time step $t - 1$. Let $P_t(\theta)^*$ denote the normalized density $P_t(\theta)$:

$$P_t(\theta)^* = \frac{e^{-\eta\lambda_{\ln}(\omega_t,\xi_t(\theta))}P_{t-1}(\theta)}{\int e^{-\eta\lambda_{\ln}(\omega_t,\xi_t(\theta))}P_{t-1}(\theta)d\theta} = \frac{\xi_t(\theta)(\omega_t)P_{t-1}(\theta)}{\int \xi_t(\theta)(\omega_t)P_{t-1}(\theta)d\theta}.$$

One can interpret $\xi_t(\theta)(\omega_t)$ as the likelihood to observe $\omega_t$ at time $t$ given the expert $\theta$. Analogously, $P_{t-1}(\theta)$ can be thought of as the prior on the experts after $t - 1$ observations. Note that crucial in this derivation is the correspondence of the exponential projection $e^{-x}$ (with $\eta = 1$) used in the definition of the pseudo-predictions and the log-loss $\lambda_{\ln}$.

## B   The Effect of $u$ on the Mixability of Losses

The crucial property making the substitution step of the Aggregating Algorithm run, is mixability. A priori it is unclear how the composition of a loss function $\lambda$ with a strictly increasing, continuous function $u$ with $u(0) = 0$ effects the mixability. We shortly argue that all interesting cases are possible. For the sake of simplicity, we look a binary outcome sets and predictions $p \in [0, 1]$. Hence, we can write the loss function as $\lambda(p) = (\lambda_0(p), \lambda_1(p))$.

(1) **non-mixable to mixable** The absolute loss $\lambda_{abs}(p) = (1 - p, p)$ is not mixable for any learning rate $\eta > 0$. Applying the function $u(x) = -\ln x$ gives $\lambda_{\ln}(p) = (-\ln 1 - p, -\ln p)$ the log-loss which is mixable, e.g., for $\eta = 0$.

(2) **mixable to non-mixable** The Brier score $\lambda_B(p) = ((1 - p)^2, p^2)$ is mixable for $\eta = 2$. However, the square root of the Brier score recovers the absolute loss $\lambda_{abs}$, which not mixable for any $\eta > 0$.

(3) **mixable to mixable** The mixable (with $\eta = 1$) log loss $\lambda_{\ln}(p) = (-\ln 1 - p, -\ln p)$ composed with $u(x) = (1 - e^x)^2$ gives the Brier score $\lambda_B(p) = ((1 - p)^2, p^2)$ which is mixable for $\eta = 2$.

## C   AA-optimality for quasi-sums

In the majority of the paper we have assumed that the prediction game in consideration is $(f, \eta)$-mixable. In this case we obtain a direct bound for the aggregated loss of the predictions given via the substitution function and the APA-QS. As it happens with the usual AA, there is no guarantee that the given loss will be $(f, \eta)$-mixable. We deal with this situation in this section. With this at hand we also provide an optimality result for the Aggregating Algorithm under quasi-sum aggregation.

First, it will be useful to impose some mild conditions on the prediction game.

**Definition C.1** (Regular Local Prediction Game (Vovk, 1995))**.** *We call the tuple $(\Omega, \Gamma, \lambda)$ a* local prediction game*. A local prediction game is called* regular *if the following four assumptions hold*

(a) *$\Gamma$ is a compact topological space.*

(b) *For each $\omega \in \Omega$ the function $\gamma \mapsto \lambda(\omega, \gamma)$ is continuous.*

(c) *There exists $\gamma \in \Gamma$ such that, $\lambda(\omega, \gamma) < \infty$ for all $\omega \in \Omega$.*

(d) *For all $\gamma \in \Gamma$ there exists $\omega \in \Omega$ such that $\lambda(\omega, \gamma) \neq 0$.*

In this section it will be always assumed that the $(\Omega, \Gamma, \lambda)$ is a local prediction game.

### C.1 Aggregation algorithm for non-mixable losses

Fix $(\Omega, \Gamma, \Theta, \lambda)$ and a weighting profile $f$, as in Section 4.3. Note, we assume that we obtain a regular local prediction game when we drop the set of experts $\Theta$. Recall that in this case, the pseudo-predictions belong to $\mathcal{P}(\lambda, f)$, that is, they are of the form

$$\psi(\omega) = g\left[\int_\Gamma f(\lambda(\omega, \gamma)Q(\gamma)\, d\gamma\right],$$

for some distribution $Q$ on $\Gamma$.

First we show that $c \geq 1$ as in (Vovk, 1990). The proof is basically the same and we include it for the reader's convenience.

**Lemma C.2.** *For given $(\Omega, \Gamma, \theta, \lambda)$ and a weighting profile $f$, then $c(f) \geq 1$.*

*Proof.* Suppose that there is $f$ such that $c := c(f) < 1$. Let $\gamma' \in \Gamma$ and $Q_{\gamma'}$ be defined as $Q_{\gamma'}(\gamma') = 1$ and 0 otherwise. Then, there exists $\gamma \in \Gamma$ such that for all $\omega \in \Omega$,

$$f(\lambda(\omega, \gamma)) \geq f\left(g\left[\int_\Gamma f(\lambda(\omega, \gamma))\, Q_{\gamma'}\, d\Gamma\right]\right)^c$$
$$\geq f(\lambda(\omega, \gamma'))^c.$$

Since $0 < f(x) \leq 1$ for all $x$, we have

$$0 \leq -\ln\left(f(\lambda(\omega, \gamma))\right) \leq c\left(-\ln\left(f(\lambda(\omega, \gamma'))\right)\right).$$

By assumption (c), we know there exists $\gamma_1 \in \Gamma$ such that $\lambda(\omega, \gamma_1) < \infty$. By the argument above (with $\gamma' = \gamma_1$) we know there is $\gamma_2 \in \Gamma$ such that

$$0 \leq -\ln\left(f(\lambda(\omega, \gamma_2))\right) \leq c\left(-\ln\left(f(\lambda(\omega, \gamma_1))\right)\right).$$

Continuing this way, we obtain a sequence $\{\gamma_k\} \subset \Gamma$ such that

$$0 \leq -\ln\left(f(\lambda(\omega, \gamma_{k+1}))\right) \leq c\left(-\ln\left(f(\lambda(\omega, \gamma_k))\right)\right).$$

Using the compactness of $\Gamma$ (assumption (a)), let $\gamma \in \Gamma$ be a limit point of this sequence. By continuity (assumption (b)), $-\ln\left(f(\lambda(\omega, \gamma))\right)$ is the limit of a subsequence $\{-\ln\left(f(\lambda(\omega, \gamma_k))\right)\}$.

Note that we have

$$0 \leq -\ln\left(f(\lambda(\omega, \gamma_k))\right) \leq c^{k-1}\left(-\ln\left(f(\lambda(\omega, \gamma_1))\right)\right),$$

thus when $k \to \infty$ we conclude that $\ln\left(f(\lambda(\omega, \gamma)) = 0$, that is $\lambda(\omega, \gamma) = 0$, which contradicts (d). □

We are now ready to obtain an analogous bound to (13).

**Theorem C.3.** *For given $(\Omega, \Gamma, \theta, \lambda)$ and a weighting profile $f$. Let $c := c(f)$. When $|\Theta| = n$ and $P_0$ is the uniform probability distribution with weights $1/n$, we have the bound*

$$\mathbf{A}_T(\text{learner}) \leq \mathbf{A}_T(\text{APA-QS}(P_0)) \leq g\left[\frac{f(\mathbf{A}_T(\theta^*))^c}{n^c}\right]. \tag{16}$$

*for any expert $\theta^* \in \Theta$.*

*Moreover, if we consider a learning rate $\eta > 0$, $f_\eta(x) = f(x)^\eta$ and $c_\eta := c(f_\eta)$, we have*

$$\mathbf{A}_T(\text{learner}) \leq \mathbf{A}_T(\text{APA}(P_0)) \leq g(f(\mathbf{A}_T(\theta^*))^{c_\eta} f(g_\eta(n^{-1}))^{c_\eta}), \tag{17}$$

*where $g_\eta$ is the inverse of $f_\eta$.*

*Proof.* Let $\psi$ be a pseudo-prediction . Then, we have $f(\lambda(\omega, \gamma) \geq f(\psi(\omega))^c$.

$$\mathbf{A}_T(\text{learner}) = g(f(\lambda(\omega_1, \gamma_1)...f(\lambda(\omega_T, \gamma_T)) \leq g(f(\psi_1(\omega_1))^c...f(\psi_T(\omega_T))^c).$$

This motivate us to define a new aggregation function given by

$$\mathbf{B}_n^c(x_1, ..., x_n) := g\left(f(x_1)^c...f(x_n)^c\right).$$

Using the fact that $c \geq 1$ (Lemma C.2) and the notation in Lemma 4.4, we see that

$$\begin{aligned}
f(\mathbf{B}^c(\theta))P_0(\theta) &= f(\lambda(\omega_1, \xi_1(\theta))^c....f(\lambda(\omega_T, \xi_T(\theta))^c P_0(\theta) \\
&\leq f(\lambda(\omega_1, \xi_1(\theta))....f(\lambda(\omega_T, \xi_T(\theta))P_0(\theta)
\end{aligned}$$

Proceeding as the in the proof of Lemma 4.4, we obtain

$$\int_\Theta f\left(\mathbf{B}_T^c(\theta)\right) P_0(\theta)\, d\theta \leq f(\psi_1(\omega_1))...f(\psi_T(\omega_T)),$$

and hence,

$$g\left[\int_\Theta f\left(\mathbf{B}_T^c(\theta)\right) P_0(\theta)\, d\theta\right] \geq g\left(f(\psi_1(\omega_1))...f(\psi_T(\omega_T))\right) = \mathbf{A}_T(\text{APA}(P_0)). \tag{18}$$

We are left to estimate the LHS of (18):

$$\begin{aligned}
g\left[\int_\Theta f\left(\mathbf{B}_T^c(\theta)\right) P_0(\theta)\, d\theta\right] &= g\left[\int_\Theta f\left(\mathbf{A}_T(\theta)\right)^c P_0(\theta)\, d\theta\right] \\
&\leq g\left[\frac{f(\mathbf{A}_T(\theta^*))^c}{n}\right] \\
&\leq g\left[\frac{f(\mathbf{A}_T(\theta^*))^c}{n^c}\right],
\end{aligned}$$

proving (16).

To obtain (17), let $f = f_\eta$ and $c_\eta := c(f_\eta)$, then we have

$$g_\eta\left[\int_\Theta f_\eta\left((\mathbf{B}^\eta)_T^{c_\eta}(\theta)\right) P_0(\theta)\, d\theta\right] = g_\eta\left[\int_\Theta f_\eta\left(\mathbf{A}_T^\eta(\theta)\right)^{c_\eta} P_0(\theta)\, d\theta\right] \leq g_\eta\left[\frac{f_\eta(\mathbf{A}_T^\eta(\theta^*))^{c_\eta}}{n^{c_\eta}}\right],$$

where $(\mathbf{B}^\eta)_n^c(x_1, ..., x_n) := g_\eta(f_\eta(x_1)^c, ..., f_\eta(x_n)) = \mathbf{B}_n^c(x_1, ..., x_n)$.

The result follows since

$$\begin{aligned}
g_\eta\left[\frac{f_\eta(\mathbf{A}_T^\eta(\theta^*))^{c_\eta}}{n^{c_\eta}}\right] &= (\mathbf{B}^\eta)_2^{c_\eta}(\mathbf{A}_T^\eta(\theta^*), g_\eta(n^{-1})) \\
&= \mathbf{B}_2^{c_\eta}(\mathbf{A}_T(\theta^*), g_\eta(n^{-1})) \\
&= g(f(\mathbf{A}_T(\theta^*))^{c_\eta} f(g_\eta(n^{-1}))^{c_\eta}).
\end{aligned}$$

$\square$

**Remark C.4.** *Recall that the weighting profile $f$ in Theorem C.3 can be written in the form $f(x) = e^{-u(x)}$ for an appropriate $u$. In this case, for $\eta > 0$, we have*

$$g(f(\mathbf{A}_T(\theta^*))^{c_\eta} f(g_\eta(n^{-1}))^{c_\eta}) = u^{-1}\left(c_\eta u(\mathbf{A}_T(\theta^*)) + c_\eta u(g_\eta(n^{-1}))\right)$$
$$= u^{-1}\left(c_\eta u(\mathbf{A}_T(\theta^*)) + \frac{c_\eta}{\eta}\ln(n)\right).$$

*In particular, when $u(x) = x$, this gives*

$$\mathbf{L}_T(\mathrm{APA}(P_0, \eta)) \leq c_\eta \mathbf{L}_T(\theta^*) + c_\eta \frac{\ln(n)}{\eta},$$

*as in (Vovk, 1990).*

## C.2 AA-optimality

Surprisingly, it is possible to show that the Aggregating Algorithm is in a game-theoretic sense optimal (Vovk, 1995). In a general game between an adversarial environment, which gets to choose experts' predictions and nature's outcome and a learner, the learner can only win if they achieve the bounds which are suggested by the Aggregating Algorithm, cf. Remark C.4. We formalize this statement and extend it to more general aggregation functions in the following.

**Definition C.5.** *Let $\mathbf{A}$ be a continuous, strictly increasing and associative aggregation function. Let $(\Omega, \Gamma, \lambda)$ be a regular local prediction game. We call the following full-information game $\mathfrak{G}$ between environment $E$ and learner $L$ a* global prediction game*:*

1. *$E$ chooses the size $n$ of a set of experts $\Theta$.*

2. *For every $t \in [T]$,*

   *(i) $E$ chooses predictions $\xi_t(\theta) \in \Gamma$ for every $\theta \in \Theta$.*
   *(ii) $L$ chooses a prediction $\gamma_t \in \Gamma$.*
   *(iii) $E$ chooses an outcome $\omega_t \in \Omega$.*
   *(iv) $\mathbf{A}_t(\theta) \coloneqq \mathbf{A}_2(\mathbf{A}_{t-1}(\theta), \lambda(\omega_t, \xi_t(\theta)))$ for all $\theta \in \Theta$.[7]*
   *(v) $\mathbf{A}_t(\text{learner}) \coloneqq \mathbf{A}_2(\mathbf{A}_{t-1}(\text{learner}), \lambda(\omega_t, \gamma_t))$.*

**Definition C.6.** *We say that the learner $L$ wins the global prediction game $\mathfrak{G}$ if for all $t \in [T]$ and $\theta \in \Theta$ there are constants $c$ and $a$ such that*

$$\mathbf{A}_t(\text{learner}) \leq u^{-1}(c\, u(\mathbf{A}_t(\theta)) + a\ln(n)), \tag{19}$$

*otherwise, we say that* nature wins*. Note, that the aggregation function is a quasi-sum with generator $u$, i.e., $\mathbf{A} = \mathbf{Q}_u$.*

Optimality here is grounded in the global game specified above. Intuitively, the following theorem shows that in a worst-case scenario, concerning the choice of experts and outcomes, every learner under expert advice can at best achieve the regret bound parametrized by $c$ and $a$ of (19). Note that we don't put any restrictions on the abilities of the learner until this point. Strikingly, the Aggregating Algorithm can achieve this regret bound, hence is optimal.

**Theorem C.7** (Optimality of Constant Regret Bound for All Predictors)**.** *Let $\mathbf{A}$ be a continuous, strictly increasing and associative aggregation function. Consider the global prediction game following Definition C.5. There exists a learner $L$ which against an arbitrary adversarial environment wins, i.e., for all $T \in \mathbb{N}$ and all $\theta \in \Theta$,*

$$\mathbf{A}_T(L) \coloneqq \mathbf{A}_T(\lambda(\omega_1, \gamma_1), \ldots, \lambda(\omega_T, \gamma_T)) \leq u^{-1}(c\, u(\mathbf{A}_T(\theta^*)) + a\ln(n)),$$

*if and only if $c \geq c(\eta)$ and $a \geq \frac{c(\eta)}{\eta}$ for some $\eta \in [0, \infty)$ with $c(\eta)$ as defined in (14), and $u$ is the generator of the aggregation, i.e., $\mathbf{A} = \mathbf{Q}_u$.*

---

[7]we set $\mathbf{A}_0(\theta) = 0$ for all $\theta \in \Theta$.

*Proof.* First, we note that the aggregation $\mathbf{A}$ fulfills (A1)-(A3). Hence, $\mathbf{A} = \mathbf{Q}^u$ for some generator $u\colon [0, \infty) \longrightarrow [0, \infty)$, continuous and strictly increasing with $u(0) = 0$ (Lemma 3.2).

Let us define the surrogate loss $\widetilde{\lambda} := u \circ \lambda$. It is straightforward to check that $(\Omega, \Gamma, \widetilde{\lambda})$ fulfills all conditions for a regular local prediction game. The first condition holds by assumption. The second condition is clear, since the composition of continuous functions is continuous. Thirdly, there exists $\gamma \in \Gamma$ such that, $\lambda(\omega, \gamma) < \infty$ for all $\omega \in \Omega$. It follows $\widetilde{\lambda}(\omega, \gamma) = u(\lambda(\omega, \gamma)) < \infty$ for all $\omega \in \Omega$. Finally, for all $\gamma \in \Gamma$ there exists $\omega \in \Omega$ such that $\lambda(\omega, \gamma) \neq 0$, hence for all $\gamma \in \Gamma$ there exists $\omega \in \Omega$ such that $\widetilde{\lambda}(\omega, \gamma) = u(\lambda(\omega, \gamma)) \neq 0$, because $u(0) = 0$ and $u$ strictly increasing. Concluding, $(\Omega, \Gamma, \widetilde{\lambda})$ is a regular local prediction game.

Theorem 1 in (Vovk, 1995) states that in the specified game the learner $L$ is guaranteed to achieve the regret bound, for all $T \in \mathbb{N}$ and all $\theta \in \Theta$,

$$\sum_{t=1}^{T} \widetilde{\lambda}(\omega_t, \gamma_t)) \leq c \sum_{t=1}^{T} \widetilde{\lambda}(\omega_t, \xi_t(\theta))) + a \ln(|\Theta|), \tag{20}$$

if and only if $c \geq \widetilde{c}(\eta)$ and $a \geq \frac{\widetilde{c}(\eta)}{\eta}$ for some $\eta \in [0, \infty)$, where

$$\widetilde{c}(\eta) := \inf\{c \in \mathbb{R} \mid \forall \psi \in \mathcal{P}(\tilde{\lambda}, e^{-\eta x}), \exists \gamma \in \Gamma, \forall \omega, e^{-\eta \tilde{\lambda}(\omega, \gamma)} \geq e^{-\eta \psi(\omega)^c}\},$$

and set $\inf\{\varnothing\} := \infty$, cf. (14). Note, that

$$\widetilde{c}(\eta) = \inf\{c \in \mathbb{R} \mid \forall \psi \in \mathcal{P}(\lambda, e^{-\eta u(x)}), \exists \gamma \in \Gamma, \forall \omega, e^{-\eta u(\lambda(\omega, \gamma))} \geq e^{-\eta u(\psi(\omega))^c}\}$$
$$= \inf\{c \in \mathbb{R} \mid \forall \psi \in \mathcal{P}(\lambda, f^{\eta}), \exists \gamma \in \Gamma, \forall \omega, f(\lambda(\omega, \gamma))^{\eta} \geq (f(\psi(\omega))^{\eta})^c\},$$

for $f = e^{-u}$, hence $\widetilde{c}(\eta)$ coincides with $c(\eta)$ defined in (14).

We give equivalent forms of (20). First, since $u^{-1}(x)$ is increasing, (20) is equivalent to

$$u^{-1}\left(\sum_{t=1}^{T} \tilde{\lambda}(\omega_t, \gamma_t))\right) \leq u^{-1}\left(c \sum_{t=1}^{T} \tilde{\lambda}(\omega_t, \xi_t(\theta))) + a \ln(|\Theta|)\right).$$

Furthermore, $\tilde{\lambda} = u \circ \lambda$ and $f(x) = e^{-u(x)}$, so (20) is equivalent to

$$\mathbf{A}_T(\lambda(\omega_1, \gamma_1), \dots, \lambda(\omega_T, \gamma_T)) \leq u^{-1}(c\, u(\mathbf{A}_T(\theta^*)) + a \ln(n)).$$

$\square$

**Remark C.8.** *The optimality of the Aggregating Algorithm under quasi-sum aggregation only refers to this specific definition of global prediction game. Note, if $c > 1$ the tightness of the regret-like bound depends on the performance of the experts. Standard $O(\sqrt{T})$-regret algorithms in learning under expert advice, e.g., Exponential Weighting Algorithm, can potentially perform better, in terms of less loss, than the Aggregating Algorithm, even though the Aggregating Algorithm is optimal in the sense specified above (Cesa-Bianchi and Lugosi, 2006, p. 14).*

# D  An illustrated, comparative example for different aggregations

Let us consider a comparative example: the simple negative utility function $u(x) = x$ corresponds to the standard sum. The negative utility function $u(x) = x^2$ generates the Euclidean norm aggregation. Compared to summation, in the Euclidean norm large loss values contribute relatively more to the result than small loss values. The analogous statement is true for the utility functions. As a negative utility function $u(x) = x^2$, large loss values hurt, since higher negative utility (cf. orange-brown arrow in Figure 2), relatively more than small loss values (cf. darkgreen arrow in Figure 2).

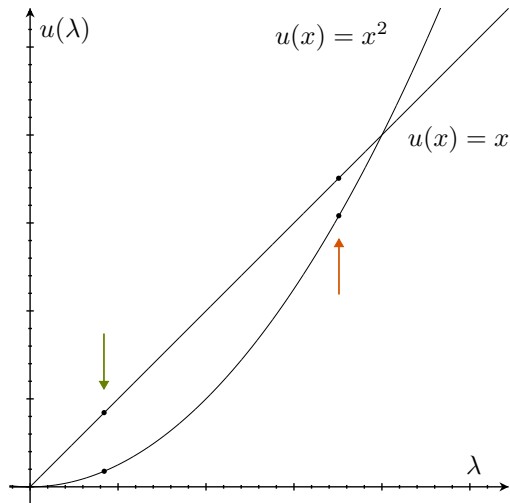

Figure 2: Comparative Example of Linear and Squared Utility. The horizontal axis denotes the loss value. The vertical axis the negative utility of the loss. We compare the negative utility function $u(x) = x$ to $u(x) = x^2$. In particular, for two values highlighted by a darkgreen arrow, low value, and an orange-brown arrow, high value.

Table 3: Weather data collections from the DWD (German Weather Agency).

| Place | File |
|---|---|
| Zugspitze | `tageswerte_KL_05792_19000801_20221231_hist.zip` |
| Potsdam | `tageswerte_KL_03987_18930101_20231231_hist.zip` |
| Putbus | `tageswerte_KL_04024_18530701_20231231_hist.zip` |
| Weissenburg-Emetzheim | `tageswerte_KL_05440_18790101_20231231_hist.zip` |

| Place | Days with Missing Values | Final Number of Days (Train/Test) |
|---|---|---|
| Zugspitze | 15 | 8386 (6708/1678) |
| Potsdam | 7 | 8759 (7007/1752) |
| Putbus | 431 | 8323 (6658/1665) |
| Weissenburg-Emetzheim | 22 | 8744 (6995/1749) |

## E   Sequential weather classification task

We run a tabular classification task on weather data collection. The data is curated and provided by the DWD (German Weather Agency). The data collections are publicly available at `https://opendata.dwd.de/climate_environment/CDC/observations_germany/climate/daily/kl/historical/`. For the file names see Table 3. Note that the files are constantly updated, hence the file names potentially change. The file names are of the format `tageswerte_KL_[location identifier]_[start date]]_[end date]_hist.zip`. Each collection provides daily measurements of weather-related attributes for one place. In every collection we deleted days with missing values. For statistic of the data collections see Table 3.

Based on daily average air pressure, average temperature, average relative humidity, maximum temperature, minimum temperature and date we train 9 classifiers to distinguish between four classes of weather: cloudy, rainy/snowy, sunny and unsettled, which are defined according to Table 4. We use the first (in chronological order) 80% of the data points as training set for the following classifiers provided by the `scikit learn` package: logistic regression (LR), gaussian naive bayes (NB), support vector machine (SVC), linear model with stochastic gradient descent (SGD), decision tree (DT), $k$-nearest neighbors (KNN), random forest (RF), bagging on decision trees (BAGGING), gradient boosting on decision trees (GB).

Table 4: Definition of weather classes.

|  | Precipitation $\leq 2mm$ | Precipitation $> 2mm$ |
|---|---|---|
| Sunshine Hours $> 4h$ | sunny | unsettled |
| Sunshine Hours $\leq 4h$ | cloudy | rainy/snowy |

Then, we run the classifiers on the remaining 20% of the data. We apply the Aggregating Algorithm for quasi-sums with the log-loss.[8] We use different aggregations in the Aggregating Algorithm as specified in Table 1. We observed that the learning rate did not have a big impact on the loss histograms in our experiment. So slightly arbitrarily, we chose $\eta = 1$ for sum and focal aggregation, we chose $\eta = 2$ for $L_{0.5}$, $\eta = 0.001$ for $L_{10}$ and $\eta = 0.5$ for $L_2$. We remark that the chosen learning rates (or the aggregations) don't necessarily guarantee that $(f, \eta)$-mixability with respect to the aggregation holds for the log-loss function. However, Theorem C.3 still applies for all aggregations.

The code was run on MacBook Pro with Intel Core i9. However, the experiments only required a fraction of the compute power. We used `Python` 3.10.2, `scikit learn` 1.4.0, `pandas` 1.4.1, `numpy` 1.22.2, `matplotlib` 3.5.1.

For the loss histograms we used automatic binning on the interval 0.0 to 3.4. Higher values, which turned out to be `np.inf`, were cut off. For this reason, we introduced the $\infty$-bars in the plots where necessary. See Section E.1 for more experiments.

---

[8]Note that the classifiers have not necessarily been trained using the log-loss function. This "mismatch" in classification tasks does not diminish the performance of the Aggregating Algorithm. However, it does allow for further optimization of the entire classification pipeline.

### E.1 More Experiments

The following tables provide more examples of the same experiment as described in Section 5.2. Note that in some of the experiments we observed $\infty$-losses due to numerical instabilities close to 0.

Table 5: Aggregations, their corresponding utility function and loss histograms for the AA-QS on the weather data collection from Potsdam. The blue line depicts the average loss excluding $\infty$-losses.

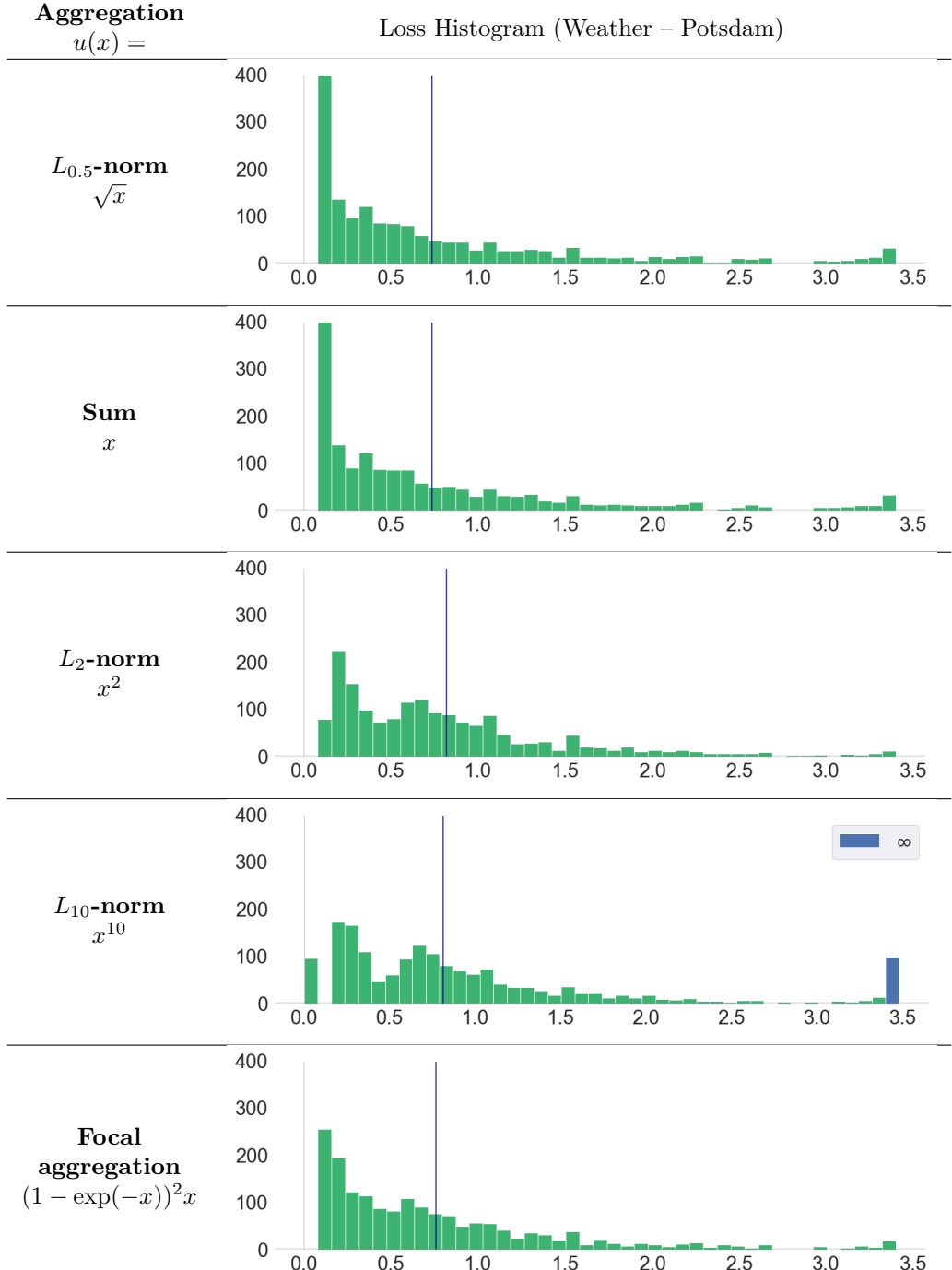

Table 6: Aggregations, their corresponding utility function and loss histograms for the AA-QS on the weather data collection from Putbus. The blue line depicts the average loss excluding $\infty$-losses.

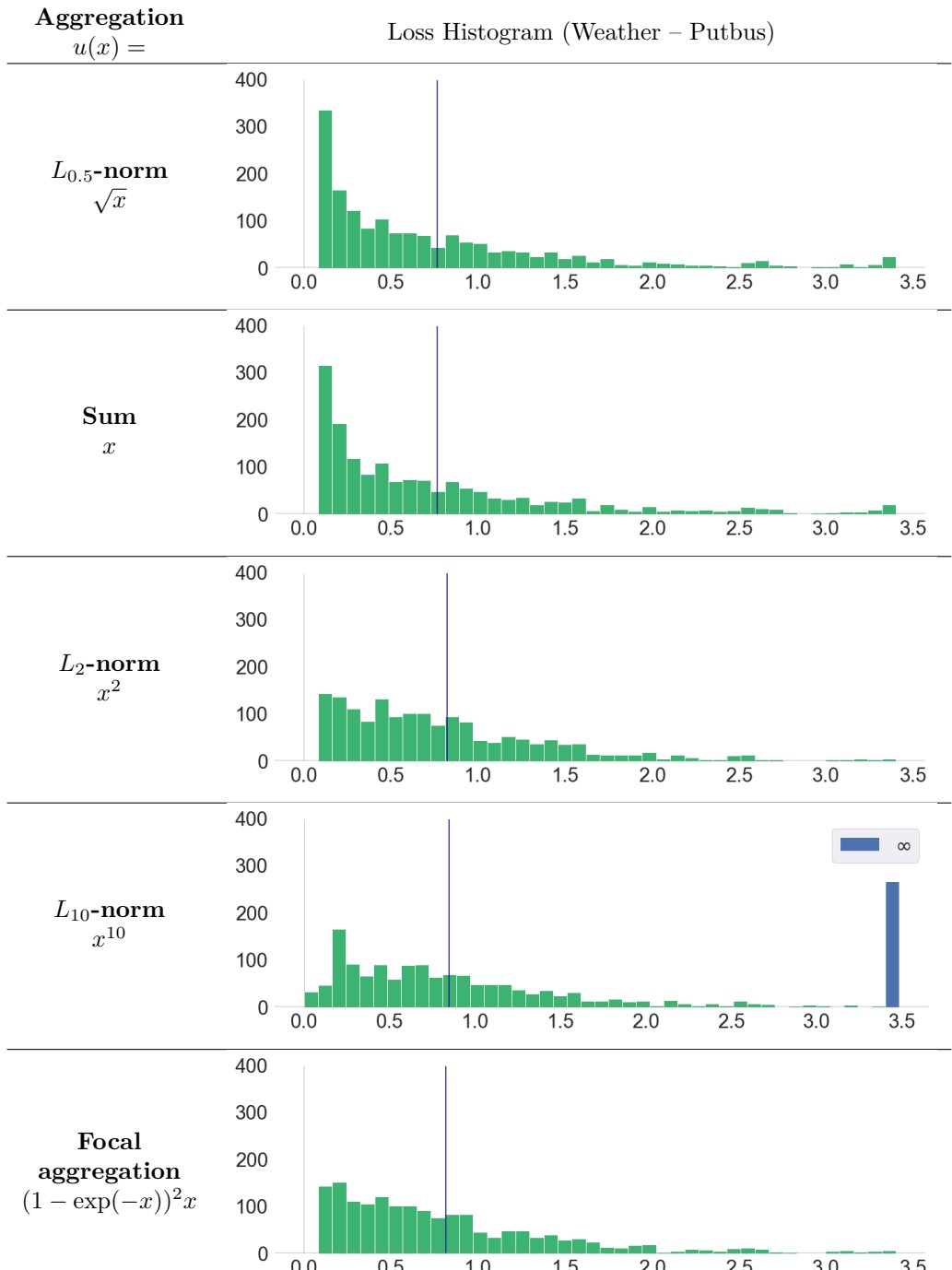

Table 7: Aggregations, their corresponding utility function and loss histogram for the AA-QS on the weather data collection from Weissenburg-Emetzheim. The blue line depicts the average loss excluding $\infty$-losses.

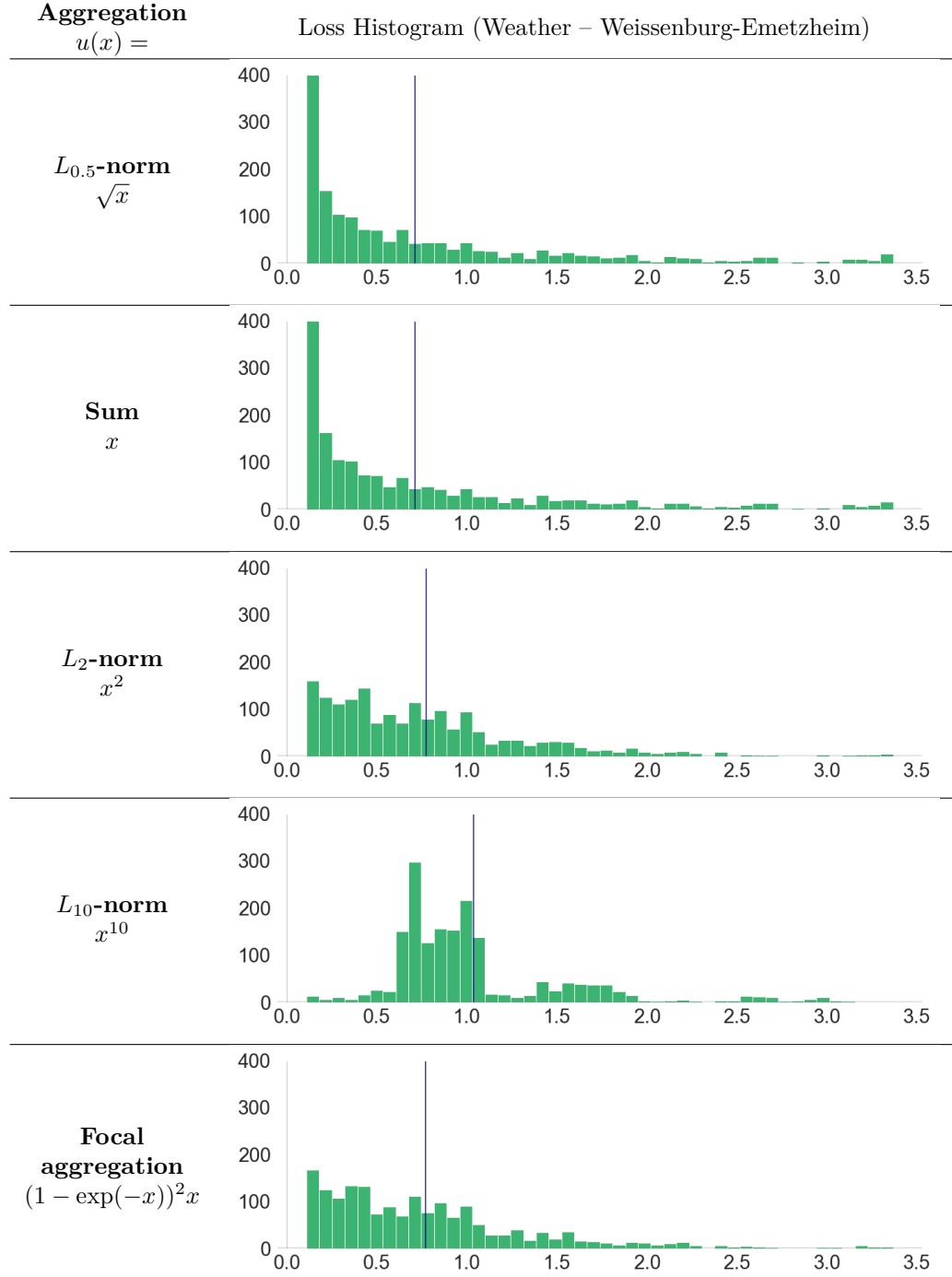

## F   Proofs

**Lemma F.1** (Proof of Lemma 3.2)**.** *Let $\mathbf{A}\colon \bigcup_{n\in\mathbb{N}}[0,\infty)^n \longrightarrow [0,\infty)$ be an aggregation function. Suppose that $\mathbf{A}$ is continuous, strictly increasing, associative and loss compatible, i.e., it satisfies* (A1) - (A4)*. Then,*

*there exists a continuous, strictly increasing function* $u \colon [0, \infty) \longrightarrow [0, \infty)$, *with* $u(0) = 0$, *such that*

$$\mathbf{A}_n(x_1, \ldots, x_n) = u^{-1} \left( \sum_{i=1}^{n} u(x_i) \right). \tag{21}$$

*If furthermore* $\mathbf{A}$ *is positively homogeneous, i.e., for every* $c \in [0, \infty)$ *and* $n \in \mathbb{N}$ *we have* $\mathbf{A}_n(cx_1, \ldots, cx_n) = c\mathbf{A}_n(x_1, \ldots, x_i, \ldots, x_n)$, *then* $u(x) = x^k$ *for some* $k \in (0, \infty)$ *in* (3).

*Proof.* Associativity (A3) guarantees that we can write

$$\mathbf{A}_n(x_1, \ldots, x_n) = \mathbf{A}_2(x_1, \mathbf{A}_2(x_2, \mathbf{A}_2(\ldots))),$$

for all $n \geq 2$ (cf. (Grabisch et al., 2009, p. 33))

In particular,

$$\mathbf{A}_3(x_1, x_2, x_3) = \mathbf{A}_2(x_1, \mathbf{A}_2(x_2, x_3)) = \mathbf{A}_2(\mathbf{A}_2(x_1, x_2), x_3).$$

Hence, $\mathbf{A}_2 \colon [0, \infty) \times [0, \infty) \longrightarrow [0, \infty)$ is monotone, i.e., strictly increasing, continuous and associative in the sense of Aczél (1948) (for an English translation see (Aczél, 2012), for a different proof (Craigen and Páles, 1989)), where it is shown that there exists $u \colon [0, \infty) \longrightarrow [0, \infty)$ strictly increasing and continuous such that

$$\mathbf{A}_2(x_1, x_2) = u^{-1} \left( u(x_1) + u(x_2) \right).$$

Since $0 = \mathbf{A}_2(0, 0) = u^{-1}(u(0) + u(0))$, it follows that $u(0) = 0$.

Finally, we obtain by induction (and associativity)

$$\begin{aligned}
\mathbf{A}_n(x_1, \ldots, x_n) &= \mathbf{A}_2(\mathbf{A}_{n-1}(x_1, \ldots, x_{n-1}), x_n) \\
&= u^{-1} \left( u \left( u^{-1} \left( \sum_{i=1}^{n-1} u(x_i) \right) \right) + u(x_n) \right) \\
&= u^{-1} \left( \sum_{i=1}^{n} u(x_i) \right).
\end{aligned}$$

For the second statement, we go back to $\mathbf{A}^2$, which is now not only strictly increasing, continuous, associative and loss compatible, but as well positive homogeneous, i.e.,

$$\mathbf{A}_2(cx_1, cx_2) = c\mathbf{A}_2(x_1, x_2),$$

for all $c \in (0, \infty)$. Hence, Theorem 2 in (Aczél, 1955) (cf. (Gardner and Kiderlen, 2018, p. 797)) applies. This implies

$$\mathbf{A}_2(x_1, x_2) = \left( x_1^k + x_2^k \right)^{\frac{1}{k}},$$

for some $k \in (0, \infty)$. By induction as above, we have

$$\mathbf{A}_n(x_1, \ldots, x_n) = \left( \sum_{i=1}^{n} x_i^k \right)^{\frac{1}{k}}.$$

$\square$

**Lemma F.2** (Proof of Lemma 3.4). *Let* $\mathbf{Q}^u$ *be a quasi-sum. Then, there exists a continuous, strictly decreasing function* $f \colon [0, \infty) \longrightarrow (0, 1]$ *with* $f(0) = 1$, *such that*

$$\mathbf{Q}_n(x_1, \ldots, x_n) = g \left( f(x_1) \ldots f(x_n) \right), \tag{22}$$

*where* $g \colon (0, 1] \longrightarrow [0, \infty)$ *is the inverse of* $f$.

*Proof.* Let $f(x) := e^{-u(x)}$. It is straightforward to check that $f$ is strictly decreasing and that $f(0) = 1$. Let $g(x) = u^{-1}(-\ln(x))$ be its inverse. Then, we can write

$$\mathbf{Q}_n(x_1, ..., x_n) = u^{-1}\left(\sum_{i=1}^n u(x_i)\right) = g\left(e^{-\left(\sum_{i=1}^n u(x_i)\right)}\right) = g\left(f(x_1)...f(x_n)\right).$$

$\square$

**Lemma F.3** (Proof of Lemma 4.4). *Let $f \colon [0, \infty) \longrightarrow \mathbb{R}$ be a weighting profile. Then*

$$\mathbf{A}_T(\mathrm{APA}(P_0)) := \mathbf{A}_T\left(\psi_1(\omega_1), \psi_2(\omega_2), ..., \psi_T(\omega_T)\right) = g\left[\int_\Theta f\left(\mathbf{A}_T(\theta)\right) P_0(\theta)\, d\theta\right].$$

*Moreover, when $|\Theta| = n$ and $P_0$ is the uniform probability distribution with weights $1/n$,*

$$\mathbf{A}_T(\mathrm{APA}(P_0)) \leq \mathbf{A}_2\left(\mathbf{A}_T(\theta^*), g\left(n^{-1}\right)\right), \tag{23}$$

*for any expert $\theta^* \in \Theta$.*

*Proof.* We will follow the general idea of the proof of Lemma 1 in (Vovk, 2001). Recall that using the updating rule (2), we have

$$P_T(\theta) = f(\lambda(\omega_T, \xi_T(\theta)))...f(\lambda(\omega_1, \xi_1(\theta)))P_0(\theta).$$

It follows that:

$$\begin{aligned}
f(\mathbf{A}_T(\theta))P_0(\theta) &= f\left(\lambda(\omega_T, \xi_T(\theta))\right) f\left(\lambda(\omega_{T-1}, \xi_{T-1}(\theta))\right) \ldots f\left(\lambda(\omega_1, \xi_1(\theta))\right) P_0(\theta) \\
&= f\left(\lambda(\omega_T, \xi_T(\theta))\right) P_{T-1}(\theta) \frac{\int_\Theta P_{T-1}(\theta)\, d\theta}{\int_\Theta P_{T-1}(\theta)\, d\theta} \\
&= \int_\Theta P_{T-1}(\theta)\, d\theta \cdot f\left(\lambda(\omega_T, \xi_T(\theta))\right) P_{T-1}^*(\theta) \\
&= \int_\Theta P_{T-1}(\theta)\, d\theta \cdot f(\psi_T(\omega_T)) f\left(\lambda(\omega_T, \xi_T(\theta))\right) f(\psi_T(\omega_T))^{-1} P_{T-1}^*(\theta) \\
&= \int_\Theta P_{T-1}(\theta)\, d\theta \cdot f(\psi_T(\omega_T)) p_T(\theta; \omega_T).
\end{aligned}$$

Integrating with respect to $\theta$ we obtain

$$\int_\Theta f\left(\mathbf{A}_T(\theta)\right) P_0(\theta)\, d\theta = \int_\Theta P_{T-1}(\theta)\, d\theta \cdot f(\psi_T(\omega_T)) \tag{24}$$

We now analyze $\int_\Theta P_{T-1}(\theta)\, d\theta$. Using similar arguments as above, we have

$$\begin{aligned}
P_{T-1}(\theta) &= f(\lambda(\omega_{T-1}, \xi_{T-1}(\theta)))f(\lambda(\omega_{T-2}, \xi_{T-2}(\theta)))...f(\lambda(\omega_1, \xi_1(\theta)))P_0(\theta) \\
&= f(\lambda(\omega_{T-1}, \xi_{T-1}(\theta)))P_{T-2}(\theta)\frac{\int_\Theta P_{T-2}(\theta)\, d\theta}{\int_\Theta P_{T-2}(\theta)\, d\theta} \\
&= \int_\Theta P_{T-2}(\theta)\, d\theta \cdot f(\psi_{T-1}(\omega_{T-1}))f(\lambda(\omega_{T-1}, \xi_{T-1}(\theta)))f(\psi_{T-1}(\omega_{T-1}))^{-1}P_{T-2}^*(\theta) \\
&= \int_\Theta P_{T-2}(\theta)\, d\theta \cdot f(\psi_{T-1}(\omega_{T-1}))p_{T-1}(\theta; \omega_{T-1}).
\end{aligned}$$

Integrating over $\Theta$ gives

$$\int_\Theta P_{T-1}(\theta)\, d\theta = \int_\Theta P_{T-2}(\theta)\, d\theta \cdot f(\psi_{T-1}(\omega_{T-1})).$$

Continuing this process we arrive to

$$\int_\Theta f\left(\mathbf{A}_T(\theta)\right) P_0(\theta)\, d\theta = \int_\Theta P_0(\theta)\, d\theta \cdot f(\psi_1(\omega_1))...f(\psi_T(\omega_T)) \tag{25}$$

$$= f(\psi_1(\omega_1))...f(\psi_T(\omega_T)) \tag{26}$$

Applying $g$ to both sides of (25), we obtain

$$g\left[\int_\Theta f\left(\mathbf{A}_T(\theta)\right) P_0(\theta)\, d\theta\right] = g\left(f(\psi_1(\omega_1))\ldots f(\psi_T(\omega_T))\right) = \mathbf{A}_T(\mathrm{APA}(P_0)),$$

as desired.

If $|\Theta| = n$ and $P_0$ is the uniform probability distribution with weights $1/n$ (cf. Vovk (1990)), we have for any fixed $\theta^* \in \Theta$,

$$
\begin{aligned}
g\left[\int_\Theta f\left(\mathbf{A}_T(\theta)\right) P_0(\theta)\, d\theta\right] &= g\left[\sum_{\theta=1}^n \frac{f\left(\mathbf{A}_T(\theta)\right)}{n}\right] \\
&\leq g\left[\frac{f\left(\mathbf{A}_T(\theta^*)\right)}{n}\right] \\
&= g\left[f\left(\mathbf{A}_T(\theta^*)\right) f\left(g\left(n^{-1}\right)\right)\right] \\
&= \mathbf{A}_2\left(\mathbf{A}_T(\theta^*), g\left(n^{-1}\right)\right).
\end{aligned}
$$

$\square$

**Corollary F.4** (Proof of Corollary 4.5)**.** *Let $f\colon [0,\infty) \longrightarrow \mathbb{R}$ be a weighting profile. Let $\eta \in (0,\infty)$ be a learning rate. When $|\Theta| = n$ and $P_0$ is the uniform probability distribution with weights $1/n$,*

$$\mathbf{A}_T(\mathrm{APA}(P_0),\eta) = \mathbf{A}_T(\mathrm{APA}(P_0)) \leq \mathbf{A}_2\left(\mathbf{A}_T(\theta^*), g_\eta\left(n^{-1}\right)\right), \tag{27}$$

*for any expert $\theta^* \in \Theta$.*

*Proof.* Consider $f_\eta(x) = f(x)^\eta$ to define $\mathbf{A}^\eta$ (see (7)), and notice that

$$\mathbf{A}^\eta(x_1,...x_n) = g_\eta(f_\eta(x_1)...f_\eta(x_n)) = g(f(x_1)...f(x_n)) = \mathbf{A}(x_1,...,x_n). \tag{28}$$

Thus, applying Lemma 4.4 with $\mathbf{A}^\eta$ we have

$$\mathbf{A}_T(\mathrm{APA}(P_0),\eta) \coloneqq \mathbf{A}_T^\eta(\mathrm{APA}(P_0)) = g_\eta\left[\int_\Theta f_\eta\left(\mathbf{A}_T^\eta(\theta)\right) P_0(\theta)\, d\theta\right].$$

Further assuming that $|\Theta| = n$ and $P_0$ is the uniform probability distribution with weights $1/n$, we have (by the proof of Lemma 4.4)

$$g_\eta\left[\int_\Theta f_\eta\left(\mathbf{A}_T^\eta(\theta)\right) P_0(\theta)\, d\theta\right] \leq \mathbf{A}_2^\eta\left(\mathbf{A}_T^\eta(\theta^*), g_\eta\left(n^{-1}\right)\right),$$

for any $\theta^* \in \Theta$.

Using (28) again, we conclude that

$$\mathbf{A}_T(\mathrm{APA}(P_0),\eta) \leq \mathbf{A}_2\left(\mathbf{A}_T(\theta^*), g_\eta\left(n^{-1}\right)\right).$$

$\square$

**Lemma F.5** (Proof of Lemma 4.7). *A loss $\lambda$ is $(f, \eta)$-mixable if and only if $\widetilde{\lambda} = u \circ \lambda$ for $u(x) := -\ln f(x)$ is $\eta$-mixable.*

*Proof.* If $\lambda$ is $(f, \eta)$-mixable, there exists a substitution function $\Sigma$ such that for all $\psi^f \in \mathcal{P}(\lambda, f_\eta)$,

$$
\begin{aligned}
& \lambda(\omega, \Sigma(\psi^f)) \le \psi^f(\omega), \forall \omega \in \Omega \\
\iff & \lambda(\omega, \Sigma(\psi^f)) \le g_\eta \left[ \int_\Gamma f_\eta(\lambda(\omega, \gamma)) Q(\gamma) \, d\gamma \right], \forall \omega \in \Omega \\
\iff & \lambda(\omega, \Sigma(\psi^f)) \le u^{-1} \left( -\frac{\ln}{\eta} \left[ \int_\Gamma e^{-\eta u(\lambda(\omega, \gamma))} Q(\gamma) \, d\gamma \right] \right), \forall \omega \in \Omega \\
\iff & u(\lambda(\omega, \Sigma'(\psi))) \le \ln_{e^{-\eta}} \left[ \int_\Gamma e^{-\eta u(\lambda(\omega, \gamma))} Q(\gamma) \, d\gamma \right], \forall \omega \in \Omega,
\end{aligned}
$$

where $\psi := u \circ \psi^f$ with $\psi \in \mathcal{P}(\widetilde{\lambda}, e^{-\eta x})$ and $\Sigma'$ is the mapping such that $\psi \mapsto \psi^f \mapsto \Sigma(\psi)$. Hence, $\Sigma$ is a substitution function for all $\psi \in \mathcal{P}(\widetilde{\lambda}, f_\eta(x) = e^{-\eta x})$, which is the standard $\eta$-mixability. $\qquad \square$

