# OpenReview forum: "Aggregating Algorithm and Axiomatic Loss Aggregation"
_TMLR — Accepted by TMLR_

### Review · Reviewer_8nXn · 2024-12-12

**Summary Of Contributions:**

This paper studies the problem of learning from expert advice by considering more general aggregation of the losses. Specifically, suppose a learner incurs losses $l_1, \cdots, l_T$. In the classical setting, the risk is simply computed as the sum $\sum_{t=1}^T l_t$, which serves as the objective for optimization. Instead, this paper considers an objective of the form $u^{-1}(\sum_{t=1}^T u(l_t))$, where $u$ is a general function. The paper shows that one can still bound the aggregated loss by running Vovk's Aggregation Algorithm on the composite loss $u \circ \lambda$, where $\lambda$ is the original loss that induces the $l_t$ values.

**Audience:**

Yes

**Claims And Evidence:**

No

**Requested Changes:**

See above.

**Strengths And Weaknesses:**

**Strengths**

1. The idea of considering a general aggregated loss seems to be original in the context of online learning.
2. The technical result in Corollary 4.1 appears to be interesting, although the proof is relatively simple.

**Weaknesses**

1. (Major) While the philosophical message of the paper is appreciated, I find the technical contribution unclear. It is not evident which result constitutes the main contribution. Is it Corollary 4.1? I recommend that the authors clearly highlight their main result in a "Theorem" environment, rather than relegating it to a lemma or corollary.

2. (Major) The technical results in Sections 4.1–4.3 appear to be restatements of Corollary 4.1. Can the authors clarify the purpose of introducing another operator $\mathbf{A}$ and function $f$, especially given that on page 9, it is mentioned that these can be converted to $\mathbf{Q}$ by setting $f = e^{-u}$? Isn't this tautological?

3. (Major) The experimental section (Section 5) is poorly presented. Can you specify the loss function $\lambda$ used and provide details on how the algorithm is implemented? Additionally, what is the purpose of studying the histogram of losses when the objectives differ (i.e., when using different $u$)?

4. (Minor) Can you instantiate your results for specific examples, such as particular choices of $u$, $\lambda$, $\Omega$, or $\Theta$? This would help readers better understand the differences compared to classical results.

5. (Minor) What are the differences between the notations $\mathbf{Q}_n^u$, $\mathbf{Q}_n$, and $\mathbf{Q}^u$? Please clarify this in the paper.


**Overall:** At this point, I am not convinced that this work represents a meaningful contribution to TMLR. However, I would be willing to reconsider my recommendation if the authors address the concerns outlined above.

---

> ### Author Response · Authors · 2025-04-23
> **Response to Weaknesses**
>
> Thanks for the review and feedback on our paper!
>
> We would like to respond to the weaknesses mentioned following the list provided by the reviewer:
>
> 1) Thanks for the comment. We do not have per se one main result that can be quoted in a theorem or lemma. Instead, we summarize our contributions on the bottom part of page 1 and top part of page 2. We will provide an explicit "contributions" identifier for this paragraph in a future version. Note that we would like to stick to the more mathematical understanding of a corollary as being a direct consequence of a theorem (which in this case is given by [Vovk, 1990]), instead of using "theorem" as a term for highlighting the importance of the statement.
>
> 2) Yes, this observation is correct. First, Corollary 4.8 and Corollary 4.1 are equivalent (we wrote: "We finally obtain Corollary 4.1 in a slightly different, but equivalent, formulation", page 9). Second, Lemma 4.4 and Corollary 4.5 look extremely related. However, these are statements about the APA, a subprocedure of the AA, which we introduce for didactical reasons and for easier comparison with Vovk's work. Note that by reiterating the argument in [Vovk, 1990], we could identify the three lessons spelled out on page 5. Regarding the notation, we use the new notation $\mathbf{A}$ and $f$ for convenience. For instance, this way we do not have to refer to $e^u$ for every use of $f$. (We write: "In the remainder of this work we will sometimes write a quasi-sum $\mathbf{Q}^u$ as an aggregation function $\mathbf{A}$ of the form (4) when convenient. This will be explicitly stated or clear from context." -> page 4). If the reviewer believes it is necessary, we could highlight this convention as a Remark.
>
>  3) Thanks! Unfortunately, we only implicitly mention the loss $\lambda$: "We provide the log-loss profile of several applications of the AA-QS for different aggregations on a sequential weather classification task." (bottom of page 11). We use the same loss for the computation of the AA-predictions. We have not explained this carefully, and we will emphasize that in a future version. Furthermore, consider Appendix E for a more detailed description of the prediction task (as referenced in this section).
>  Note that we study the different histograms to better understand the influence of the aggregation function (which yes, it does change the objective function) and in this case nicely highlights the different effects on which losses will be incurred. The purpose of the figure is to compare the incurred loss values by the AA depending on the choice of aggregation. That is, the pictures should not be equal because then the aggregation would not have any influence on the prediction made by the AA.
>
>  4) We are not entirely sure about the motivation of the question. In principle, we could instantiate the results for specific choices. Is the reviewer asking to provide Corollary 4.1 in terms of a specific choice of $u, \lambda, \Omega, \Theta$? If the reviewer is interested in getting an intuitive introduction to the AA algorithm for a specific instantiation, we refer to Appendix A (as we do on page 11).
>
>  5) Thanks for pointing out: in fact $\mathbf{Q}_n$ was a typo in Lemma 3.4, thanks for catching it! We use the index $n$ to make a difference between a general aggregation function and an aggregation of $n$ instances (see Definition 3.1).

---

### Review · Reviewer_TTUp · 2025-01-03

**Summary Of Contributions:**

This paper proposes a novel approach to generalizing loss aggregation functions through the use of quasi-sums, integrating them into the well-established Aggregating Algorithm (AA) framework.

While the method essentially adapts AA to a new loss function, the axiomatic characterization of loss aggregation functions and the derivation of regret bounds for quasi-sums are both well-justified and mathematically rigorous.

The paper is mostly well-organized, and examples like weather prediction help illustrate the theoretical concepts.

**Audience:**

Yes

**Broader Impact Concerns:**

N/A/

**Claims And Evidence:**

Yes

**Requested Changes:**

#### **Critical Adjustments**

1. **Motivation and Background**
   - Expand the introduction to clearly articulate the significance of the problem being addressed. Specifically:
     - Justify why alternative loss aggregation methods are necessary in online learning compared to traditional risk minimization approaches.
     - Explain the importance of transitioning from offline to online settings and why this generalization matters for practice.
     - Address the abrupt transition to "learning under expert advice" and provide context for its relevance.
   - Add more background on online learning to help readers understand why new loss functions, such as quasi-sums, are beneficial.

2. **Discussion of Related Work**
   - Provide a detailed comparison with existing works, particularly the **tilted empirical risk minimization** framework ([1]) and other relevant methods.
     - Discuss similarities, differences, and complementary aspects of these approaches.
     - Explain the advantages of the proposed framework over these methods, particularly in terms of regret bounds and applicability to online learning.

3. **Clarity of Theoretical Contributions**
   - Elaborate on the differences between the axiomatic approaches to loss aggregation in offline and online learning, as implied in Section 3.1. This distinction is crucial for understanding the novelty and significance of the proposed method.
   - Provide more intuitive explanations or diagrams for dense sections, such as the generalization of mixability constants.

4. **Experimental Validation**
   - Expand the empirical study to include multiple datasets and tasks, especially those with diverse loss distributions and real-world challenges.
   - Perform a more rigorous analysis of the experimental results, including statistical evaluations to substantiate claims about learner attitudes (e.g., risk-averse vs. risk-seeking behaviors).

#### **Non-Critical Adjustments**

1. **Computational Implications**
   - Address the computational bottlenecks of the substitution step, especially in high-dimensional settings. While not critical, proposing strategies to mitigate these issues would enhance the practical applicability of the method.

2. **Empirical Comparisons**
   - Include empirical comparisons with other online learning methods, such as the weighted majority algorithm, to provide a clearer picture of the framework's performance.

**Strengths And Weaknesses:**

### **Strengths**

1. **Novelty and Rigor**
   The paper introduces an innovative axiomatic framework for generalizing loss aggregation functions in online learning, expanding beyond traditional summation methods. The theoretical contributions, including the characterization of quasi-sums and the derivation of regret bounds, are mathematically rigorous and demonstrate originality.

2. **Clarity of Examples**
   Including examples, such as the weather prediction task, effectively bridges the gap between theoretical concepts and their practical implications. These examples help clarify the proposed approach and illustrate its potential utility in real-world scenarios.

3. **Expansion of the Aggregating Algorithm (AA) Framework**
   The paper broadens its applicability to a wider range of learning scenarios by systematically integrating quasi-sum aggregation functions into the AA framework. This added flexibility makes adapting the AA for different loss structures possible, enhancing its practical relevance.

4. **Potential for Practical Insights**
   The framework introduces the idea of weighting profiles and utility functions, offering a new perspective on learner attitudes (e.g., risk aversion or risk-seeking behavior). This interpretive layer adds depth and could inspire further research on tailoring online learning algorithms to specific applications.

---

### **Weaknesses**

1. **Lack of Motivation**
   The introduction does not clearly explain why the problem addressed by this paper is important. For instance:
   - The abrupt statement *"However, an axiomatical approach to generalized loss aggregations remained within the borders of offline learning"* lacks context on the necessity of extending such approaches to online learning.
   - The importance of "learning under expert advice" is mentioned without explaining its relevance in the broader context of online learning challenges.
   A stronger narrative explaining the practical and theoretical importance of this work would significantly improve its impact.

2. **Insufficient Discussion of Related Work**
   The paper does not adequately position itself within the existing literature. For example, the tilted empirical risk minimization framework (the following [1]) offers a related approach to replacing traditional summation, but the connection to this work is not discussed.

3. **Limited Background and Context**
   The paper would benefit from a more detailed explanation of online learning, its inherent challenges, and why alternative loss aggregation functions are needed. A robust background section would help readers understand the significance of quasi-sums and their potential impact.

4. **Narrow Experimental Scope**
   The empirical validation relies solely on a weather prediction dataset, which is insufficient to demonstrate the generalizability of the approach. Broader experiments across diverse tasks and datasets would:
   - Highlight the practical relevance of the proposed framework.
   - Provide stronger evidence for the claims about the impact of different aggregation functions on learner behavior.

5. **Computational Challenges**
   The substitution step of the Aggregating Algorithm introduces a potential computational bottleneck, particularly in high-dimensional settings. While this issue is mentioned, the paper does not propose strategies to address it, which limits its practical applicability in complex tasks.

---

### **References**

[1] Li T, Beirami A, Sanjabi M, et al. *On tilted losses in machine learning: Theory and applications*. Journal of Machine Learning Research, 2023, 24(142): 1-79.

---

> ### Author Response · Authors · 2025-04-23
> **Response to Critical Weaknesses**
>
> Thanks for the review and feedback on our paper! We appreciate the concrete suggestions made by the reviewer.
>
> We respond to the critical adjustments (which mainly follow the weaknesses given by the reviewer) following the list provided by the reviewer:
>
> 1) Thanks for pointing this out and let us respond in several comments. Unlike offline learning, online learning does not require any distributional assumptions on the realized outcomes $y$. This is particularly helpful in scenarios including agents, which react to predictions where the assumption of a stable distribution is likely to be false. The large literature with seminal books like [Cesa-Bianchi and Lugosi, 2006, Prediction, Learning, and Games] emphasizes the relevance. Many contributions at COLT embrace the paradigm of not assuming any underlying distribution. Note that learning under expert advice is a central problem setup in online learning. Now, to emphasize why is it relevant to look at generalizations of loss aggregation in online learning compared to offline learning, in offline learning, we are interested in the average loss, as the number of instances should not influence the comparison of "goodness". In online learning, it is more common to consider summed up losses, because this allows to provide more fine-grained statements on the regret behavior. For example, mixable losses give constant regret, while more general convex losses only give $\log n$ regret bounds in learning under expert advice. Hence, we consider an axiomatic approach to summed up losses and quasi-sums. For averages, analogous approaches exist, e.g. [Fröhlich et al., 2024, Risk measures and upper probabilities], or Theorem 4.10 in [Grabisch et al., 2009, Aggregation Functions].
>
> Anyway, we do see the necessity of extending the motivation of an axiomatic approach (as this has also been mentioned by another referee). One simple motivation is that an axiomatic approach allows debating and discussing the/a "right" set of axioms required for a certain scenario in online learning. For instance, "associativity" is a strong assumption and does not allow to express a time-dependent importance of certain instances versus others. We consider an axiomatic approach as a reductive way to understand the core building blocks of a certain object relevant to the problem. In this case, the aggregation of losses. The axiomatization demarcates a certain area for which, in this case the AA, provides regret guarantees.
>
> 2) In [Li et al, 2023, On tilted losses in machine learning: Theory and applications] the authors suggest replacing the standard ERM objective by a different risk minimization objective. The average is replaced by a "quasi-average" using a family of exponential functions. Such quasi-averages for more general generators have been part of studies on aggregation functions, see Section 4.2 in [Grabisch et al., 2009]. In particular, the presented quasi-means (in their book called "quasi-arithmetic mean") generalize the empirical Renyi divergences from [Li et al., 2023]. Note that in particular, Kolmogorov and Nagumo have proven an analogous axiomatical characterization to our Lemma 3.2 (this is referenced in the text, page 4). Hence, the objective in [Li et al., 2023] is a special case of an average analogue of our quasi-sums. We consider quasi-sums because online learning literature usually focuses on sums and not averages, as a fine-grained dependency is more important to express (as argued above). We will add a more detailed related work section in a future version and thank the referee for pointing to [Li et al., 2023]. What other relevant methods do you refer to? Note that we do not see how the TERM framework can be directly applied to online learning.
>
> 3) The main difference is the use of a normalizer, in other words, the difference between average and sum. For sum-type aggregations we are not aware of any axiomatic approach to loss aggregation. For offline learning with average losses there is [Fröhlich et al., 2023]. Again, we argue (as above) that summation of losses is crucial to obtain fine-grained regret bounds in online learning. Furthermore, we wanted to ask whether Appendix A helps to clarify the meaning of "mixability"? And which section do you refer to with "such as the generalization of mixability constants"? Section 4.3? We acknowledge that some of the results are technical and their proofs not entirely simple. We hope that with Appendix A we provide sufficient intuition to understand the results. However, this might not necessarily constitute an intuitive step-by-step guide to the proofs of the statements.
>
> 4) We do not see good reasons for running more experiments. In particular, we got the feedback by another reviewer that the "experiments wouldn't even be necessary" after all. Nevertheless, we are happy that the experiments shed a bit more light on the presented results.

---

> > ### Author Response · Authors · 2025-04-23
> > **Response to Non-Critical Weaknesses**
> >
> > 1) That is a different but interesting question. Potentially, our analysis will help to get closer to an answer, since the "change of variables" in the AA gives a way to transform the problem into a potentially "easier" problem. Note that there are many technical and mathematical challenges in trying to make this precise (for example, the potential loss of mixability), which go far away from the scope of this work.
> >
> > 2) We are not sure how an empirical comparison to other methods contributes to our analysis. Note that on a theoretical level, the AA (even under general aggregations) obtains a constant regret bound, which, for instance, the weighted majority algorithm does not.

---

> > > ### Comment · Reviewer_TTUp · 2025-05-28
> > > **Thanks for the responses**
> > >
> > > Thank you for the detailed and thoughtful feedback. I appreciate the clarifications regarding the motivation and theoretical background. The explanation of why summation-based aggregation is more appropriate in online settings, and how it enables fine-grained regret analysis, clearly addresses my earlier concerns.
> > >
> > > The discussion on the relationship with the tilted empirical risk minimization (TERM) framework and quasi-arithmetic means was also helpful. It clarified how your work generalizes and differentiates from existing approaches. I now better understand that quasi-sums offer a meaningful extension for online learning scenarios where average-based methods may not be directly applicable.
> > >
> > > I also found Appendix A particularly helpful for understanding the technical aspects, especially the concept of mixability and how it is extended in your setting.
> > >
> > > Regarding the experiments: I understand your position and agree that, given the theoretical nature of the paper, extensive empirical evaluation is not essential. If the primary contribution lies in the axiomatic development and the connection between quasi-sums and utility-based modeling (as also noted by Reviewer hCiS), then the current experiments serve a sufficient illustrative purpose.
> > >
> > > Overall, the clarifications have addressed my concerns, and I appreciate your responsiveness to the feedback.

---

### Review · Reviewer_hCiS · 2025-04-15

**Summary Of Contributions:**

This paper proposes a generalization of Vovk's Aggregating Algorithm in the context of online learning with expert advice. The key contribution is the introduction of quasi-sum aggregation functions of the form $$Q_u(x_1,\ldots,x_n)=u^{-1}\left(\sum_iu(x_i)\right),$$ which generalizes the usual additive loss aggregation used by the classic aggregating algorithm. A generalized form of mixability is introduced to handle non-linear aggregations of the losses. The analysis captures the existing guarantees as a special case, while enabling more general expressions of a learner's risk attitude.

**Audience:**

Yes

**Claims And Evidence:**

Yes

**Requested Changes:**

My main requests would be a more thorough discussion of the related prior works mentioned in the previous section, some additional discussion about the possible limitations of the axioms explored here (e.g. inability to capture temporal locality), and some additional motivation for why an axiomatic perspective is inherently valuable or desirable

**Strengths And Weaknesses:**

**Strengths**

The generalization of the classic aggregating algorithm seems like it could be useful for studying risk-sensitive problem settings, or for designing robust experts algorithms. The generalized mixability condition could potentially be very useful, given that regular mixability is already quite useful.

**Weaknesses**

- I feel that the paper fails to properly situate itself in relation to existing literature. The base idea behind these quasi-sums is strongly related to many existing notions in game theory and economics that I feel these aren't properly discussed. For instance, much of section 5 seemed a bit redundant to me --- many of the interesting implications of these quasi-sums are already well understood and well studied through the lens of utility theory. This connection is hinted at briefly in section 5 but not systematically explored to a meaningful degree. Indeed, if this connection was thoroughly explored the experiments wouldn't even be necessary: it is already well-understood from utility theory how convex and concave utility functions relate to risk aversion/seeking. There's no reason to explore these questions empirically as there's already quite a broad literature on the topic, answering precisely these questions. Moreover, note that the utility theoretic perspective of preferences is inherently axiomatic as well, dating back to the seminal work of Von Neumann & Morgenstern (1947). Similarly, there are connections to the wide literature on risk-sensitive learning that are similarly axiomatic in nature, particularly wrt the prospect theoretic point of view and the study of coherent risk measures such as the Conditional Value at Risk.

- While paper presents a nice set of generalized results, it would benefit from more discussion of the implications of this generalization. For instance, how might the properties of particular utility functions effect convergence rates or learning stability.

- Section 3 states that each of the axioms are natural to impose, but I'm not sure I agree that the associativity axiom is all that natural for an online learner. Aggregation strategies which capture temporal locality will generally not satisfy this axiom. For instance, any weighted averaging or discounting schemes will generally not be associative. Yet handling "local averages" of this kind would be important to express attitudes related to temporal structure (e.g. learners which adapt to a non-stationary data sequence need to "forget" old data over time in order to track a time-varying solution). If such a fundamental consideration doesn't fit into the current set of axioms it makes me question whether these really are the "right" axioms for an online problem setting.

- I'm not entirely sure I understand the value of developing an axiomatic characterization. I can glean bits and pieces from the exposition here (e.g., remark 4.10 discussing how the log-loss being seemingly fundamental for summation aggregation but not necessarily other types of aggregation). However, overall I felt that this motivation was not very well articulated throughout the paper, and I came away without a clear vision of why I should want to start from axioms rather than just starting from the performance measure that I care about (e.g., regret, weighted regret, etc), as one usually would in online learning.

- I found the general layout and development of the paper a bit puzzling. It felt like many versions of the same results were being repeated with only minor variations (e.g. Corollaries 4.1, 4.5, 4.8, Lemma 4.4, all seem like they're all showing minor variations of the same result).

**Questions**
- Is the utility of these quasi-sums firmly affixed to the LEA setting, or do you expect that the techniques here could be similarly applied to reason about different ways of aggregating losses in OLO? i.e. FTRL can be viewed as doing a sort of loss aggregation, by setting $w_{t+1}=\nabla\psi^*(-\sum_{s=1}^tg_s)$, which is related to regret in terms of cumulative loss.
- Are there any interesting implications for strongly-adaptive/dynamic regret? For instance, the fixed-share algorithm is able to achieve nice regret guarantees on every sub-interval when considering mixable losses (see e.g. the second edition of Hazan's text). Perhaps something interesting could be said by considering more general aggregations.
- Since the goal is to develop an axiomatic view, can it be shown that the set of proposed axioms is a minimal characterization of these quasi-sums?
- What is the inherent value of developing an axiomatic approach?

**Minor Typos**
- p5/6: "why does a loss distortion can be translated into a change of aggregation", "on this way", "For the sake of the moment"
- eq (2) the logarithm is seemingly in base $e^{-\eta}$?

---

> ### Author Response · Authors · 2025-04-23
> **Response to Weaknesses**
>
> We thank the reviewer for the detailed feedback!
>
> 1) We agree that Section 5 recovers what has been known about utility functions. Nevertheless, we find it helpful to illustrate the effect of utility functions in the context of online learning. We have not, to the best of our knowledge, seen this before. The link to the axiomatic representation results in decision theory, in particular, [von Neumann and Morgenstern, 1944/1947], is worth mentioning. Thanks for pointing this out, we will revise this in a future version. Unfortunately, we are not aware of a direct relationship between the suggested axioms for aggregation and the axioms for a preference relationship to be representable by a utility function. We believe this could be an interesting future research question.
> We don't directly see how prospect theory [Wakker, 2008, Prospect Theory for Risk and Ambiguity] relates to our axiomatical approach which is not motivated by describing decision behavior.
>
> In terms of other axiomatical characterizations of aggregations, it is worth mentioning coherent risk measures (e.g. such as CVar) (We refer to [Fröhlich et al., 2023], which relates to coherent risk measures in finance, e.g. [Delbaen, 2002].), but as well [Kolmogorov et al., 1933] (which we cite as well). The latter is an axiomatic characterization of quasi-averages (analogous to our Lemma 3.2), which could be related to von Neumann and Morgenstern's characterization. However, we don't know of any work explaining the exact relationship. Closest to this question we found [Roberts, 1980, Interpersonal Comparability and Social Choice Theory]. However, the goals and thus the axioms are quite different, even though the resulting aggregation is very similar to CVar. As well, [Zen, 1974, Informational Bases of Alternative Welfare Approaches] has a clear connection of utility and aggregation. But the axioms motivated by social welfare considerations which make ours distinct from theirs.
>
> Another central difference between the name axiomatical approaches to aggregations is that generalizations of averages normalize (with respect to the number of instances), while we are interested in un-normalized aggregations, as this is necessary for the fine-grained analysis of regret bounds in online learning.
>
>
> 2) We are not sure whether we understand correctly. Do you refer to the computation of the substitution function? Can you elaborate? Note that the regret guarantees persist (up to constants) for difference choices of $u$.
>
> 3) It is correct that the choice of axioms is and should not be universal. In particular, as you point out, the associativity axiom is a debatable one. It does a lot of heavy lifting for the axiomatical characterization. In a future version of the article we will weaken the statement "natural" to "dependent on context, the axioms seem relevant". We were interested in finding more helpful axioms but we have not made any progress on this. However, we are convinced that there are other sets of axioms which could be of interest in online learning.
>
> 4) There is one very natural and direct motivation for an axiomatic characterization which is: one can discuss the axioms. As you reasonably questioned the associativity axiom before, the set of axioms provides a nice starting point for discussions. Furthermore, a set of axioms provides a list of potentially desirable or undesirable properties of an aggregation which then can systemically ruled out or be adapted. In addition, our specific choice of axioms emphasize the special role of quasi-sums as they allow for the switch from additive to multiplicative structure necessary for the AA. We will include those motivations for axiomatically studying loss aggregations in a future version.
>
> 5) The similarity of the results is not surprising as we re-analyze the AA in other terms. Note that 4.1 and 4.8 are equivalent (as mentioned) and that 4.4 and 4.5 are statements about APA and not AA (which look extremely similar to 4.8 which is stated for AA). We have chosen the structure because we wanted to go through the claims listed on page 5 bottom and page 2 top. We start with 4.1 as the observation that a change of variables is sufficient to make the AA work for quasi-sums is the starting point of the re-analysis of the AA.

---

> > ### Author Response · Authors · 2025-04-23
> > **Response to Questions**
> >
> > a) We do think that a generalized approach to loss aggregation via quasi-sums is transferable to other relevant methods (in LEA) such as FTRL. In particular, a related change of variables for the loss could be possible here. However, we have not gone through the details. The question is interesting and potentially part of future work.
> >
> > b)  We don’t have any particular insights for strongly-adaptive/dynamic regret setups. In principle, we don’t see any problem to generalize the aggregation functional in dynamic regret. However, we are unsure whether in this case we would have to adapt the path-length dependent bounds as well, which usually are a summation of the differences of the comparison predictions. That is, the utility function $u$ might change over time. These are wild guesses and we would be happy for a more detailed input.
> >
> > c) That is a fine question. We cannot directly see whether the set of axioms is minimal or not. From our understanding, this would require a significant amount of work, which is beyond the scope of this project and our goals.
> >
> > d) Because we can discuss and critique the choice of axioms. Axiomatizations provide a small number of statements revealing an object's mathematical structure. The axiomatic approach to loss aggregation has been proven useful in offline learning already (cf. [Fröhlich et al., 2023]). Plus, more generally, it is central to other related topics of machine learning such as decision theory [von Neumann and Morgenstern, 1944/1947] and information theory [Shannon, 1948].

---

> > ### Comment · Reviewer_hCiS · 2025-04-23
> >
> > My apologies, I had meant to include some references at the end
> > of my review but forgot to paste them in:
> > - Prospect theory
> >   - Kahneman, Daniel, and Amos Tversky. "Prospect theory: An analysis of decision under risk." Econometrica 47.2 (1979): 363-391.
> >   - Tversky, Amos, and Daniel Kahneman. "Advances in prospect theory: Cumulative representation of uncertainty." Journal of Risk and uncertainty 5 (1992): 297-323.
> > - Coherent risk measures / risk-sensitive learning
> >   - Artzner, Philippe, et al. "Coherent measures of risk." Mathematical finance 9.3 (1999): 203-228.
> >   - Bäuerle, Nicole, and Ulrich Rieder. "More risk-sensitive Markov decision processes." Mathematics of Operations Research 39.1 (2014): 105-120.
> >
> > The connection to Prospect theory I was referencing actually dates back much further than the 2008 Wakker paper referenced in the
> > response, and it very much is an axiomatic approach. Moreover, both VaR and CVaR are captured by
> > the framework of prospect theory as special cases, so if we agree that CVaR should be discussed then we ought
> > to also agree that the more general prospect theory should also be discussed.
> >
> >
> > > There is one very natural and direct motivation for an axiomatic characterization which is: one can discuss the axioms.
> >
> > This does not seem so convincing to me; when I pointed out an axiom that does not seem to be very natural for an online learning context,
> > the response seemed to just boil down to "we chose these ones, and they're convenient for the analysis". In general
> > I think these discussions will boil down to agreeing to disagree, since the choice of which axioms are reasonable is inherently subjective.
> > Instead, when starting from the usual objective-based approach, it is clear which design decisions lead to good outcomes: the ones
> > which optimize some objective the designer is interested in.
> >
> > > 2. We are not sure whether we understand correctly...
> >
> > I meant more along the lines of consequences of choosing a particular utility function on the algorithm's performance, so something like an example
> > where choosing a particular aggregation function would enable improvements of some kind. For instance, how in a non-stationary setting
> > choosing a discounting aggregation (if it were a valid choice given the axioms) would enable tracking a time-varying solution.
> > On the other hand, perhaps there are situations where a poor choice of aggregation could cause the algorithm to diverge/incur linear regret.

---

> ### Author Response · Authors · 2025-04-24
> **Response to Response I**
>
> NOTE: Because of the character limit we respond in two parts.
>
> Thanks a lot for specifying the references in more detail! And thanks for the fast and detailed response.
>
> We have to apologize in making a mistake in the previous response:
> [Kolmogorov, 1933] was actually meant to be [Kolmogorov, 1930, Sur la notion de la moyenne]. To be fair, Nagumo independently discovered the same axiomatizational characterization in [Nagumo, 1930, Über eine Klasse der Mittelwerte].
>
> In case, some of the references were unclear:
> - [Delbaen, 2002] -> [Delbaen, 2002, Coherent risk measures on general probability spaces]
> - [Shannon, 1948] -> [Shannon, 1948, A mathematical theory of communication]
> - [von Neumann and Morgenstern, 1944/1947] -> [von Neumann and Morgenstern, 1944/1947, Theory of games and economic behaviour]
> - [Fröhlich et al., 2023] -> [Fröhlich et al., 2023, Risk Measures and Upper Probabilities: Coherence and Stratification]
>
>
> 1)
> To the first comment of the reviewer: We are sorry to make the impression that prospect theory seems of less relevance to us because we claim it would not be an axiomatized theory. What we wanted to say is that we think that the comparison to prospect theory is less relevant because the motivation for the axioms is different: describing human decision behavior versus identifying central (debatable) rules for aggregating in online learning.
>
> This ties back to the comment on Var and CVar being part of prospect theory. Let us focus on CVar. The axiomatical characterization (Theorem 1) referred to in [Tversky and Kahneman, 1992, Advances in prospect theory: Cumulative representation of uncertainty] states that a preference relationship on prospects fulfilling certain axioms is equivalently representable by a split up Choquet integral over gains and losses.
>
> Very closely related to cumulative prospect theory there is rank dependent expected utility theory [Appendix B.4, Fröhlich et al., 2023, Risk Measures and Upper Probabilities: Coherence and Stratification]. The main difference is that cumulative prospect theory makes a distinction between losses and gains while rank dependent expected utility theory considers absolute values (Tversky and Kahneman call rank dependent expected utility theory "cumulative utility theory" in Note 2 (last page) in [Tversky and Kahneman, 1992, Advances in prospect theory: Cumulative representation of uncertainty]). Now, rank dependent expected utility theory states that a preference relationship on acts fulfilling certain axioms is equivalently representable by a Choquet integral with capacities being distorted probabilities (e.g. [Yaari, 1987, The dual theory of choice under risk]). If this distortion is concave, then the capacity is submodular (page 19 [Fröhlich et al., 2023, Risk Measures and Upper Probabilities: Coherence and Stratification]]), which makes the Choquet integral a coherent risk measure. In particular, CVar is a parametric family of such coherent risk measures.
>
> On the other hand, there are direct axiomatizations of aggregations e.g. for coherent risk measures (e.g. [Delbaen, 2002, Coherent risk measures on general probability spaces] or [Artzner et al., 1990, Coherent measures of risk]. Regarding the connection between axiomatisation of preferences (à la von Neumann and Morgenstern) and axiomatisation of aggregation functionals, we are aware (unpublished work by one of the authors) that one can derive a bi-implication between certain axiom sets on preferences and certain axiom sets on aggregation functionals. But we do not see this as helpful for our present purpose because we do not think that arguing in terms of preference orders is as intuitive as directly looking at the properties of the aggregation functional as we have done.
>
>
>
> 2)
> We entirely agree to the reviewer in that the choice of axioms is somewhat subjective. Nevertheless, the axioms provide a basis for discussion, though.
>
> Consider the objective-based approach: at the beginning of the design pipeline relevant stakeholders try to define or converge to an objective. Hence, a discussion of the relevant parameters, choices (for instance of loss function or aggregation) is required. How do the persons find an answer? By structuring the existing options of, e.g., losses and aggregations. Given axiomatical characterizations of aggregations certain properties of aggregations can be discussed, accepted as being important or neglected as being irrelevant.
> Clearly, the special axiomatization we came up with can only be a first step and does not lay out the entire option space of aggregations in online learning.

---

> > ### Author Response · Authors · 2025-04-24
> > **Response to Response II**
> >
> > 3)
> > Thanks a lot for the clarification! That is a good question.
> > In our case, an example of this approach would be: what is the consequence of using an algorithm which optimizes for standard summed up loss, instead for some distinct quasi-sum aggregated loss, on a setup on which an algorithm optimizing for this quasi-sum aggregated loss would perform well? For instance, let us consider a pool of experts in which one expert (call it A) is consistently slightly off in its predictions while another expert (call it B) is predicting perfectly, just sometimes it makes a huge mistake (in terms of loss). Then, one would expect that an AA with a risk-averse aggregation profile would tend to value the predictions by the expert A more than expert B (in terms of: the predictions made by AA would be somewhat closer to A then to B). On the other hand, if the AA is not risk-averse, but risk-seeking, it would ``prefer'' agent A over B.
> > We have made experiments on this in artificial scenarios and have observed this behavior under certain circumstances. We didn't include the results, because we were not convinced by the significance as it was hard to clean the setups such that we could exclude the case that the observation of the behavior wasn't a simple artefact of the artificial scenario. On the theoretical side we made several attempts to better understand the influence of the aggregation on the prediction behavior, without success.
> >
> > There is a logical problem however in trying to compare two different aggregation schemes on a given problem to see which is better: the aggregation is an essential part of the overall evaluation measure. We are not proposing different algorithms for a given objective: we are actually changing the objective. The question would be like choosing different utility functions in an economic problem and asking which is “best”. The point is the utility is supposed to model something in the world. Likewise, the aggregation method is modeling the overall means by which a prediction method is judged.
> >
> > For this reason, we understand our main contributions as providing a first hint that an axiomatic characterization of loss aggregations in online learning can be helpful, in this case as it allows to understand the components and their semantics in the AA.

---

> > > ### Comment · Reviewer_hCiS · 2025-08-13
> > > **Post-revision update**
> > >
> > > I have read the revisions posted to the paper; thank you for the detailed update of the manuscript.
> > >
> > > I am somewhat unconvinced still by revisions. While the new revisions acknowledge the weaknesses discussed, I don't feel that some of the primary concerns were actually addressed in a substantial way. For instance, the existence of similar classic axiomizations is swept under the rug without much justification:
> > >
> > >  > The axiomatic characterizations of aggregations for loss functions should not be conflated with axiomatic characterizations of decision behavior. Certainly, there are relations between the axiomatic characterization of expected utility behavior based on the seminal work by Von Neumann and Morgenstern (1953) and the axiomatic characterization of quasi-arithmetic means by Kolmogorov and Castelnuovo (1930) and Nagumo (1930) mentioned before (Muliere and Parmigiani, 1993). However, the goals of the axiomatic statement and hence the axioms are quite different.
> > >
> > > I don't really see how this addresses the concern: the point of both characterizations is that different aggregations correspond to different objectives. The characterization in terms of decision behaviour simply phrases this fact in terms of a learner optimizing that objective, but this doesn't seem like a fundamental distinction between those works and what is presented here. Moreover, while it is claimed that the axioms are quite different, it is also acknowledged that they are functionally the same as those from the classic results:
> > >
> > > > The set of quasi-arithmetic means forms the set of idempotent analogues to the set of quasi-sums character- ized above (see (Grabisch et al., 2009, Section 6.5.1) for details). Kolmogorov and Castelnuovo (1930) and Nagumo (1930) independently provided an axiomatic characterization of this set which strongly resembles our Lemma 3.2.
> > >
> > > Could you comment a bit further on these details?

---

> > > > ### Author Response · Authors · 2025-08-14
> > > > **Response to Post-Revision Update**
> > > >
> > > > Thanks for considering the updates and giving more feedback!
> > > >
> > > > We have to admit that we are not sure whether we understand you correctly.
> > > >
> > > > In our understanding, characterisation results in decision theory are of the following type: given an axiomatically described preference structure between acts, the preference structure could equivalently be represented by a certain type of aggregation over outcomes (e.g., probability distribution over outcomes) and a utility function. Characterisation results in aggregation theory are of the type: given an axiomatically described aggregation, the aggregation can be described by certain parametrisation, e.g., in our case a utility function.
> > > >
> > > > We would be very happy to resolve our confusion. For instance, you could potentially provide formal (mathematical) statement, or even perhaps an example, which makes the characterisations “closer” in their nature? (This is how we understand your claim.)
> > > >
> > > >
> > > > To your second point: when we say “the goals of the axiomatic statement and hence the axioms are quite different”, we mean the goals of the axiomatic statement for decision theory versus the goals of the axiomatic statement for aggregations (in the case of the referred paragraph, the quasi-arithmetic means).
> > > >
> > > > Additional comment on idempotency: we certainly acknowledge resemblance with previous work, which is due to the analogy of idempotent (e.g., quasi-arithmetic means) and non-idempotent aggregation functions (e.g., ours). However, it is not the case that if one is given an axiomatic description of an idempotent aggregation, the axiomatic description of the non-idempotent analogue is readily given. For instance, we are not aware of an axiomatic description of the non-idempotent analogue of coherent risk measures. But, coherent risk measures are idempotent and axiomatically described. Since this seems to be part of the points that left you unconvinced,
> > > >
> > > > Could you please be specific about what is the concern about having some resemblance with classical results? The proofs are certainly different and, to our knowledge, do not appear in the literature. Moreover, they extend our understanding of non-idempotent aggregation functions and their relation to online learning (via the Aggregation Algorithm).

---

> > > > > ### Comment · Reviewer_hCiS · 2025-08-16
> > > > >
> > > > > I think the connection is more fundamental than is being implied. The point of studying loss aggregation is that different aggregations of the losses correspond to different objectives. Put another way, different aggregations will lead to different
> > > > > "preferences" or "risk attitudes" in any agent
> > > > > optimizing that objective, which is precisely what is studied in utility theory (but phrased in terms of "maximizing utility" instead
> > > > > of "minimizing loss").
> > > > > It does not seem sufficient to write these off as simply "axiomatic descriptions of behaviour": they are studying the
> > > > > same thing as this manuscript, but with different phrasing.
> > > > >
> > > > > The axioms of Von Neumann and Morgenstern can not be
> > > > > written off as merely characterizations of behaviour, they are a characterization of utility functions (See e.g. Chapter 3.6 of
> > > > > Von Neumann & Morgenstern 1953), and are further generalized to different weightings (ie. aggregations) of utilities
> > > > > by cumulative prospect theory (CPT) (Kahneman & Tversky (1979), Tversky & Kahneman (1992)).
> > > > > Indeed, the main objects in CPT are the utility function (loss function)
> > > > > and the weighting function (aggregation scheme). Moreover, prospect theory and cumulative prospect theory
> > > > > *are* axiomatic characterizations of measures of risk, and capture the coherent risk measures of artzner et al. 1999
> > > > > as special cases, so it is not completely clear to me that the axiomatic characterization of the coherent risk measures via
> > > > > the aggregating algorithm necessarily extends what was already known, or to what extent.
> > > > >
> > > > > To summarize, my point was that while the updated manuscript now acknowledges
> > > > > that there is some overlap between these ideas, it seems to significantly downplay how
> > > > > fundamental this relationship really is, writing these off as "axiomatic descriptions of behavior"
> > > > > as if it's completely orthogonal, when they are actually very closely related and relevant to what's being
> > > > > studied here. The setting studied in the paper can be interpreted as a particular
> > > > > subset of utility theory, and the loss aggregation scheme here even seems to potentially be
> > > > > less general than the value functions described by CPT, so Section 3.1 of the manuscript seems
> > > > > misleading to me.
> > > > >
> > > > > Additionally, Section 3.1 also dismisses Neyman and Roughgarden (2023)
> > > > > as studying aggregations of experts rather than aggregations of losses, but
> > > > > these seem to be equivalent under the quasi-arithmetic pooling studied there (Section 3.6).
> > > > >
> > > > > > Could you please be specific about what is the concern about having some resemblance with classical results
> > > > >
> > > > > The concern is that the degree to which they are related is not discussed in sufficient depth, and it is
> > > > > not clear from what is presented what is genuinely novel and could not be done in the existing
> > > > > frameworks from utility theory. The specific choice of axiomization in terms of aggregation functions
> > > > > is new and necessitates
> > > > > different proofs from existing literature, but which insights generated from this
> > > > > set up are genuinely new and couldn't otherwise be obtained using the classical
> > > > > utility theoretic frameworks?